# Laser-induced fluorescence lidar detection of weak biomass burning aerosols at Nanping, South China

Zhekai Li<sup>1,2</sup>, Dawei Tang<sup>3</sup>, Tianwen Wei<sup>1,2</sup>, Saifen Yu<sup>1,2</sup>, Jing Cai<sup>1,2</sup>, Kenan Wu<sup>4</sup>, Zhen Zhang<sup>1,2</sup>, Jiadong Hu<sup>1,2,5</sup>, Haobin Han<sup>1,2,5</sup>, Yubin Wang<sup>1,2,5</sup>, and Haiyun Xia<sup>1,2,5,6</sup>

Correspondence: Tianwen Wei (twwei@nuist.edu.cn) and Haiyun Xia (hsia@ustc.edu.cn)

Abstract. South China, a densely populated region frequently affected by transported biomass burning aerosols (BBA), is in need of sensitive remote sensing observations to characterize these plumes. Laser-induced fluorescence (LIF) lidar is powerful tool for detecting fluorescence aerosols and has recently been demonstrated to identify transported BBA over Europe, while its applications in South China remain scarce. Here, we present LIF lidar observations of fluorescent aerosols conducted at Nanping, South China. The detected fluorescence layer exhibited relatively weak intensity (maximum fluorescence backscatter coefficient ≈ 0.16 × 10<sup>-5</sup> Mm<sup>-1</sup> Sr<sup>-1</sup> nm<sup>-1</sup>), more than two orders of magnitude lower than the N₂ Raman backscatter signal. Nevertheless, it showed a distinct spectral signature compared with typical urban aerosols. By integrating multi-source datasets, the fluorescence layer was attributed to long-range transported BBA originating from weak fire activity in the Indo-China Peninsula (ICP). Furthermore, the concurrent presence of BBA and enhanced water vapor indicated a humid environment favorable for aerosol processing. This study demonstrates that multi-channel LIF lidar provides a sensitive and promising approach for detecting and characterizing BBA layers over South China, thereby offering new insights into their transport mechanisms and potential environmental impacts.

<sup>&</sup>lt;sup>1</sup>State Key Laboratory of Climate System Prediction and Risk Management, Nanjing University of Information Science and Technology (NUIST), Nanjing 210044, China

<sup>&</sup>lt;sup>2</sup>School of Atmospheric Physics, NUIST, Nanjing 210044, China

<sup>&</sup>lt;sup>3</sup>Academy of Chips Technology, China Electronics Technology Group Corporation, Chongqing 401332, China

<sup>&</sup>lt;sup>4</sup>School of Information Engineering, Huangshan University, Huangshan 245041, China

<sup>&</sup>lt;sup>5</sup>Institute of Lidar Technology, GuangZai Co., Ltd., Hangzhou 310005, China

<sup>&</sup>lt;sup>6</sup>School of Earth and Space Science, University of Science and Technology of China, Hefei 230026, China

30

## 1 Introduction

Biomass burning aerosols (BBA), predominantly emitted from wildfires, have significant impacts on atmospheric processes and public health. BBA can warm the atmospheric layer in which they reside and play an important role in aerosol–cloud–precipitation interactions (Lin et al., 2014). Once emitted, they undergo complex chemical transformations during transport, which affect downwind air quality and atmospheric composition (Zhou et al., 2017). Among BBA components, biomass burning organic aerosols are of particular concern due to their persistence in the atmosphere and potential health risks (Majdi et al., 2019; Vasilakopoulou et al., 2023). Interactions between wildfires, smoke, and meteorological conditions can also form positive feedback loops that aggravate regional air pollution and associated health outcomes (Huang et al., 2023).

Indo-China Peninsula (ICP), a sub-region of Southeast Asia, is a major source of BBA during the pre-monsoon dry season and has substantial impacts on regional air quality and global climate (Yadav et al., 2017). Several international campaigns have investigated BBA in this region, including BASE-ASIA (Biomass-burning Aerosols in South-East Asia: Smoke Impact Assessment) and 7-SEAS (Seven South-East Asian Studies) (Lin et al., 2013). Multiple studies have reported the meteorological conditions during the dry pre-monsoon months, particularly March and April, promote vertical lifting of BBA over the ICP. These elevated plumes are subsequently transported eastward under the influence of prevailing monsoonal winds, reaching South China and even the Western Pacific, and can produce severe health effects (Chi et al., 2010; Chang et al., 2013; Heese et al., 2017; Nguyen et al., 2020; Qin et al., 2024; Xue et al., 2025; Liu et al., 2025). Therefore, sensitive remote sensing observations are needed in these downwind regions.

In early 2005, Immler et al. reported an unexpected enhancement in the water vapor Raman channel signal, which they attributed to fluorescence interference from BBA (Immler et al., 2005). Since then, researchers began to develop the single-channel (Rao et al., 2018; Veselovskii et al., 2021; Zhang et al., 2021) and multi-channel (Sugimoto et al., 2012; Saito et al., 2018; Richardson et al., 2019; Reichardt et al., 2023; Wang et al., 2023; Huang et al., 2025; Tang et al., 2025a) laser-induced fluorescence (LIF) lidar systems, which offer enhanced capabilities for environmental applications. Previous BBA observations using LIF lidar have been conducted primarily in Europe, where fluorescent layers and characteristic BBA spectra were reported; those layers were often associated with strong fires in North America or Russia. (Reichardt, Jens et al., 2018; Hu et al., 2022; Veselovskii et al., 2023; Gast et al., 2025; Veselovskii et al., 2025). However, fluorescence spectra of BBA can vary substantially across locations and cases (Reichardt et al., 2025). Consequently, further observations in different locations and weaker cases are needed, particularly in areas with high biomass burning emission potential and population density such as the ICP and South China, where LIF remote sensing observations remain limited.

Building on these advances, we conducted LIF lidar observations at Nanping, South China, during April and May 2024, and detected a distinct fluorescence layer. Although the fluorescence intensity was relatively low, the layer exhibited spectral signatures that different from those of urban aerosols. By integrating multi-source datasets, we attributed this fluorescence layer to BBA transported from weak fire sources in the ICP. Radiosonde and LIF lidar data further suggested that the BBA layer was transported together with elevated water vapor, indicating a humid transport pathway.

## 2 Observations and data

## 2.1 Multi-channel LIF lidar

The LIF lidar was installed on the top floor of the National Center of Carbon Metrology Building in Nanping City, Fujian Province, China (26.59° N, 118.27° E). The system emitted 355 nm lasers from an Nd:YAG laser (Innolas Spitlight EVOIII) with energy greater than 200 mJ; the laser was then expanded by a 10× beam expander, and finally redirected to the atmosphere at an elevation angle of 30° via a reflector. The echo signal was gathered by a 12-inch telescope (Meade LX 200) and detected by the lidar detector equipped with a 32-channel photomultiplier tube (Licel SP32HR). A 355 nm optical notch filter was placed between the telescope and lidar detector to suppress elastic signal leakage. Many types of atmospheric aerosols can produce fluorescence under UV laser excitation, particularly organic carbon (OC), a major component of BBA. The dominant fluorescence emission wavelength occurs within 400–650 nm when excited by a 351 nm laser (Pan, 2015). To include both Raman and fluorescence signals simultaneously (Table 1), in this study, the spectrometer's central wavelength was set to 475 nm, with a spectral resolution of 6.2 nm mm<sup>-1</sup>. Accordingly, the effective detection ranges of the 32 channels (labeled Channels 0–31) span from 378.9 nm to 571.1 nm. Four observation cases (Case 1–4) in April and May 2024 were discussed. Among them, a distinct BBA layer was identified in Case 1, while the remaining cases served as comparisons. Further technical details of the LIF lidar system are available in our previous publication (Tang et al., 2025a).

Table 1. Channel settings of different detection signals.

| Content                 | Central wavelength (nm) | Channel index |
|-------------------------|-------------------------|---------------|
| N <sub>2</sub> Raman    | 385.1                   | 30            |
| H <sub>2</sub> O Raman  | 403.7                   | 27            |
| N <sub>2</sub> Overtone | 422.3                   | 24            |
| Fluorescence            | 434.7–571.1             | 22–0          |

60

## 2.2 Satellite and weather observations

The Moderate-Resolution Imaging Spectroradiometer (MODIS) is a spaceborne multispectral sensor widely used in BBA studies (Shi et al., 2019; Yin, 2020). In this study, we utilized the MODIS Corrected Reflectance (True Color) imagery acquired on April 16, 2024, to visually overview fire activities in the ICP. We selected two sub-regions with dense fires from the imagery to illustrate the spatial distribution of fire sources (Fig. 5a–b). To further quantify and identify fire events, we obtained the MODIS Collection 6.1 (C6.1) standard active fire product from NASA's Fire Information for Resource Management System (FIRMS) (Giglio et al., 2016). This dataset includes fire locations, fire radiative power (FRP), detection confidence, and fire types. For quality control, we used fire points with confidence > 80%, FRP > 100 MW, and classified as type 0 (presumed vegetation fires).

The AErosol RObotic NETwork (AERONET) (Holben et al., 1998) is a global ground-based aerosol monitoring program jointly established by NASA and PHOTONS (a European initiative coordinated by the University of Lille 1, the French National Centre for Space Studies, and the National Institute for Earth Sciences and Astronomy of CNRS). To support the identification of BBA transport events, we selected three AERONET sites near burning areas in the ICP (Chiang\_Mai\_Met\_Sta, Doi\_Ang\_Khang, and Luang\_Namtha); their geographic locations are marked as red rectangles in Fig. 5e. Temporal variations in the aerosol optical depth at 500 nm (AOD<sub>500</sub>) provided auxiliary evidence for biomass burning plumes. We obtained air pollution data (including PM<sub>2.5</sub> and PM<sub>10</sub> concentrations) from the China National Environmental Monitoring Centre (CNEMC). To reduce localized variability and measurement noise, we averaged data from three monitoring stations near the LIF lidar site. For the water vapor mixing ratio (WVMR) profiles, we used sounding data of five cities in South China downloaded from the University of Wyoming's website; the location distributions are shown in Fig. 7a. To avoid cloud contamination in the sounding profiles, we discarded any level with relative humidity (RH) exceeding 95%, following the approach of Zhou et al. (Zhou et al., 2025).

## 2.3 Reanalysis data and model

ERA5 (Hersbach et al., 2020) is the fifth-generation global climate reanalysis dataset from the European Centre for Medium-Range Weather Forecasts (ECMWF). In this study, we used ERA5 data to characterize boundary layer height, wind fields, and RH at different altitudes, providing essential meteorological context for our analysis. The Modern-Era Retrospective analysis for Research and Applications, Version 2 (MERRA-2) is a long-term global reanalysis developed by NASA's Global Modeling and Assimilation Office (GMAO). It assimilates satellite-based aerosol observations to represent interactions between aerosols and other physical processes in the climate system (Gelaro et al., 2017). In this study, we used hourly biomass burning OC emissions from the MERRA-2 tayg1\_2d\_adg\_Nx hourly dataset with a spatial resolution of 0.625×0.5 degrees.

The Hybrid Single-Particle Lagrangian Integrated Trajectory (HYSPLIT) model, developed by NOAA's Air Resources Laboratory, is widely used to analyze atmospheric transport, dispersion, air parcel trajectories, and pollutant transport pathways (Stein et al., 2015). In this study, we employed HYSPLIT to compute 78 hours backward trajectories from the LIF lidar site, aiming to identify the transport history and potential source regions of the observed fluorescent aerosol layer. The meteorological inputs were from the Global Data Assimilation System (GDAS).

## 95 3 Calibration and retrieval of LIF lidar

The calibration procedure was initially performed on the original fluorescence spectrum. Additionally, unavoidable imperfections in the diffraction grating can cause periodic spectral artifacts in fluorescence spectra, especially under weak fluorescence conditions. In our study, minor spurious peaks were found at channels 21, 16, and 10 (Fig. 1). These peaks, known as Lyman ghosts (Tang et al., 2025a), were corrected based on their dependence on the primary wavelength (N<sub>2</sub> Raman signal; channel 30, with a central wavelength of 385.1 nm) (Meggers and Kiess, 1922). In Fig. 1, the spectra are normalized by the N<sub>2</sub> Raman signal. For each case, both uncorrected and corrected curves are shown for direct comparison. To quantify ghost contributions

**Figure 1.** Mean fluorescence spectra for Case 1, Case 2 (1.0–1.8 km altitude), and Case 4 (2.0–2.8 km altitude) measured at the Nanping lidar station. Line colors indicate the different cases; solid lines show spectra before ghost-line correction and dashed lines show spectra after ghost-line correction (see legend). All the spectra are normalized by the N<sub>2</sub> Raman signal.

and standardize correction across all spectra, we selected the spectrum with the lowest fluorescence intensity during the campaign (the Case 4 uncorrected curve, bottom blue solid line in Fig. 1). For this spectrum, the three ghost-affected intervals (channels 21, 16, and 10) were removed and reconstructed using linear interpolation, yielding a ghost-free reference spectrum (bottom blue dashed line in Fig. 1). The differences between the two blue curves at the three affected channels represent the characteristic ghost contribution for each channel. Ghost-correction coefficients were then defined as these channel-specific differences divided by the corresponding  $N_2$  Raman signal intensity from the same spectrum. All analyzed spectra used the same three coefficients (one per ghost-affected channel). During the correction, ghost contributions were estimated by multiplying the  $N_2$  Raman signal intensity of each affected channel by its corresponding coefficient, and these were subtracted from the original values to yield the corrected spectra. Representative results for Case 1 and Case 2 are shown in Fig. 1, where the dashed lines denote ghost-free spectra.

The aerosol extinction coefficient is given by (Ansmann et al., 1990):

$$D = d/dz \left\{ \ln \left[ N_{R}(z) / P_{R}(z) z^{2} \right] \right\}$$

$$\tag{1}$$

$$\alpha_{\rm L}^{\rm aero}(z) = \frac{D - \alpha_{\rm L}^{\rm mole}(z) - \alpha_{\rm R}^{\rm mole}(z)}{1 + (\lambda_{\rm L}/\lambda_{\rm R})^k} \tag{2}$$

where the superscripts "aero" and "mole" are quantities related to aerosols and molecules, respectively. The subscripts "L" and "R" correspond to elastic and  $N_2$  Raman channels.  $N_R$  is the nitrogen concentration profile, which is calculated using the standard atmosphere model (Shang et al., 2018).  $\lambda_L$  and  $\lambda_R$  denote the elastic wavelength (355 nm) and the  $N_2$  Raman channel central wavelength (385.1 nm), respectively. The Ångström exponent k is assumed to be 1 (Ansmann et al., 1992).

The fluorescence backscattering coefficient is given by (Veselovskii et al., 2020):

$$\beta_{\rm F} = \frac{C_{\rm R}}{C_{\rm F}} \frac{P_{\rm F}}{P_{\rm R}} \frac{T_{\rm R}}{T_{\rm F}} N_{\rm R} \sigma_{\rm R} \tag{3}$$

where  $P_{\rm F}$  and  $P_{\rm R}$  are the lidar signals from the fluorescence and N<sub>2</sub> Raman channels, respectively.  $\sigma_{\rm R}$  is the N<sub>2</sub> Raman differential backscattering cross section at 355 nm (Venable et al., 2011).  $T_{\rm R}$  and  $T_{\rm F}$  are the atmospheric transmittance for the N<sub>2</sub> Raman and fluorescence (channel 0–22, with wavelengths of 434.7–571.1 nm), respectively. Similarly, the water vapor Raman backscattering coefficient  $\beta_{\rm H_2O}$  is derived by replacing the fluorescence channels with the water vapor Raman channel (channel 27, with a central wavelength of 403.7 nm). Unlike the single-channel fluorescence systems, our LIF lidar uses 23 fluorescence channels. So the total fluorescence backscattering coefficient is obtained by summing contributions from all individual channels. For comparison, the resulting  $\beta_{\rm F}$  is normalized by the fluorescence spectrum's wavelength range:

$$\overline{\beta}_{F} = \frac{\beta_{F}}{\Delta \lambda} \tag{4}$$

## 4 Results

## 130 4.1 Vertical profiles observed by LIF lidar

Time-height profiles of four representative cases observed in 2024 are shown in Fig. 2. All observations were conducted at night to avoid strong solar background interference during the daytime. Four cases are included: Case 1 (19 April, 23:19–20 April, 00:05), when a distinct fluorescence layer was observed; Case 2 (11 May, 01:41–02:37); Case 3 (18 May, 21:43–23:29); and Case 4 (23 May, 21:27–23:08). All times are in local time (UTC+8), with the time axis labeled at 20 minute intervals. The vertical white lines separate individual cases. Fig. 2a-c show the range corrected signal (RCS) of different detection channels, while Fig. 2d–f present retrieved parameters derived via the methods described in Sect. 3. Due to the LIF lidar's 30° elevation angle, the effective detection height equals half of the line-of-sight range. The detection lower limit is within 800 m, which defines the starting altitude of all subplots. The N<sub>2</sub> Raman signal (Fig. 2a) decays more slowly in Case 4 than in the other three cases, likely due to the lower aerosol loading (Fig. 2d and Fig. A1a). The fluorescence signal is also relatively weak in Case 4 (Fig. 2c, f, and Fig. 3), making it ideal for calibrating the Lyman ghost lines. Multiple cloud layers are identifiable via high  $\alpha_{\rm L}^{\rm aero}(z)$  values in Fig. 2d, consistent with observations reported by Sugimoto et al. In several cases, low-level clouds exist around 3.5 km, significantly attenuating the lidar signal. During the observation period, both  $\overline{\beta}_{\rm F}$  and  $\alpha_{\rm L}^{\rm aero}$  (0.8–1.4 km layer, Fig. 2d and f) closely match the local PM concentration trends (Fig. A1a). Their corresponding average fluorescence spectra also exhibit similar shapes (Fig. 4a), consistent with previous observations of urban aerosols (Veselovskii et al., 2025).

**Figure 2.** Time-height profiles for the four cases at the Nanping lidar station during 19 April–23 May 2024 (UTC+8). (a–c) The range corrected signal (RCS) from the respective detection channels (see Table 1 for channel definitions). (d–f) The retrieved quantities (derived using the methods described in Sect. 3) corresponding to (a–c), respectively.

These lines of evidence support that low-altitude fluorescence is largely attributable to urban aerosols. In Case 3, an elevated boundary layer height (Fig. A1b) indicates enhanced vertical mixing. The low-altitude fluorescence layer ascends to ~3 km. This upward shift is further evident in Fig. 3, where the β̄<sub>F</sub> in the low-altitude layers decays rapidly in all cases except Case 3. In Case 1, a distinct fluorescence layer accompanied by enhanced water vapor was observed at ~1.8 km despite relatively low α<sub>L</sub> (Fig. 2c, e, and f and Fig. 3). Such fluorescence enhancement is absent in the other three cases (Fig. 3). We attributed the
1.8–2.4 km layer in Case 1 to BBA transported from the ICP. Detailed source-attribution results are presented in Sect. 4.3.

# 4.2 Fluorescence spectra of BBA

As shown in Fig. 4, mean fluorescence spectra from multiple 600 m thick layers are normalized to the  $N_2$  Raman signal. The box colors are consistent with the layers marked in Fig. 3. The maximum fluorescence intensity is more than two orders of magnitude lower than that of the  $N_2$  Raman signal. Although the overall fluorescence intensity is weaker than reported in previous Asian LIF lidar studies (Sugimoto et al., 2012; Wang et al., 2023), clearly discernible spectral patterns can be classified into two distinct groups. Spectra in Fig. 4b correspond to urban aerosols, showing a gradual intensity decrease with increasing wavelength. This is consistent with urban aerosol spectra in the boundary layer reported in Russia (Veselovskii et al.,

Figure 3. Mean vertical profiles of  $\beta_{\rm H_2O}$  (blue lines) and  $\overline{\beta}_{\rm F}$  (red lines) for the four cases at the Nanping lidar station. Error bars indicate  $\pm$  1 standard deviation across the profile samples. Shaded boxes mark the 600 m thick atmospheric layers; box colors denote aerosol types (yellow for BBA; gray for urban aerosols).

2025). One layer in Case 2 (1.8–2.4 km, shown in Fig. 2c and f and Fig. 3), although located above the local boundary layer, matches this urban spectral pattern and is therefore classified as such. By contrast, the BBA layer (1.8–2.4 km layer in Case 1) exhibits stronger fluorescence with distinct peaks. Notably, the low-altitude layer (0.8–1.4 km) in Case 1 shows no distinct fluorescence maximum in the time-height profiles (Fig. 3) and would therefore be expected to exhibit a typical urban aerosol spectrum. Instead, its mean spectrum exhibits a weak peak closely resembling the higher-altitude (1.8–2.4 km) BBA signature.

## 4.3 Source attribution of the fluorescence layer in Case 1

HYSPLIT backward trajectories for Case 1 indicate that the air mass linked to the fluorescent layer originated from the ICP during 16–18 April, 2024 (Fig. 5c). This event can be divided into three stages: local biomass burning, vertical lifting, and long-range transport. MODIS true-color imagery for 16–17 April (Fig. 5a–b) reveals multiple fire points, with near-surface plume orientations consistent with winds blowing toward the northeast (Fig. 5f–g). Two source regions with the most intense fire activity are highlighted by black boxes in Fig. 5d–g (these boxes correspond to the areas as in Fig. 5a–b). MODIS C6.1 fire products also identify numerous low-intensity vegetation fire pixels within these regions (Fig. 5d), while the MERRA-2 emission fields indicate elevated OC emissions coincident with these suspected sources (Fig. 5f–g). Additionally, the 10-m wind barbs align with the plume directions visible in the satellite imagery. As shown in Fig. 6, AOD<sub>500</sub> at three AERONET stations (locations marked as red rectangles in Fig. 5e) increased over a period, indicating the passage of strongly absorbing aerosols. The BBA then underwent vertical lifting to a higher altitude (Fig. 5c). During the pre-monsoon season, frequent small-scale cumulus convection can transport pollutants above 3 km–an optimal altitude for long-range transport (Shan et al.,

**Figure 4.** Mean fluorescence spectra (normalized by N<sub>2</sub> Raman signal) for the 600 m thick layers indicated in Fig. 3. Line colors and marker shapes indicate the cases and layer altitudes (see legend). Box colors correspond to aerosol types defined in Fig. 3 (yellow for BBA; gray for urban aerosols). (a) Spectra for layers influenced by BBA. (b) Spectra corresponding to urban aerosols.

2016). For our case, a similar phenomenon was visible in Fig. 5a–b. Additionally, the higher terrain (Fig. 5d) also provides favorable conditions for BBA uplift. Thereafter, the BBA were rapidly transported by the southwest summer monsoon (Fig. 5d) to altitudes above our LIF lidar site, where a fluorescent layer was observed. All of the above lines of evidence indicate that BBA generated by the fire points were lifted, transported to South China, and eventually detected by our LIF lidar in Case 1. On the contrary, HYSPLIT trajectories results with a starting altitude of 1 km show that the low-altitude fluorescence was not affected by the transported BBA (Fig. B1).

**Figure 5.** Imagery, trajectory analysis and modeled biomass burning emissions for Case 1. (a–b) MODIS (Aqua) true-color images acquired on 16 April, 2024 (UTC), sourced from NASA Worldview (https://worldview.earthdata.nasa.gov/), showing smoke plumes originating from multiple fire points (areas highlighted by black boxes in panels d–g). (c) Vertical variation of air mass altitude from HYSPLIT backward trajectories (starting altitude: 2.1 km; black triangle marks the Nanping lidar station). (d) Horizontal (longitude-latitude) HYSPLIT backward trajectories overlaid on presumed vegetation fire locations; marker colors and shapes indicate FRP levels, and the underlying digital elevation model (DEM) highlights topography (the black triangle denotes the lidar station). (e) Locations of the AERONET stations (red triangles) used in Fig. 6. (f–g) MERRA-2 biomass burning organic carbon (OC) emissions (Kg m<sup>-2</sup> s<sup>-1</sup>) for two temporal conditions, plotted with background 10-m wind fields.

**Figure 6.** Aerosol optical depth at 500 nm (AOD<sub>500</sub>) retrieved from three AERONET sites during 16–22 April 2024. Marker colors and shapes indicate the stations: CM represents Chiang\_Mai\_Met\_Sta, DAK represents Doi\_Ang\_Khang, and LN represents Luang\_Namtha (locations shown in Fig. 5e, red triangles).

## 5 Discussion

200

The Asian summer monsoon brings substantial rainfall to the region. However, the pre-monsoon season (February–April) is typically dry across the ICP. During this period, biomass burning events (predominantly forest fires) occur frequently over the ICP, driving seasonal maximum BBA loadings (Gautam et al., 2013; Shi and Yamaguchi, 2014). Fire activity peaks in March but declines sharply by late April (Huang et al., 2016). MODIS FRP data indicate that over 84% of source fires had FRP < 500 MW (with only a few high-intensity events), while MERRA-2 data show a maximum OC emission rate of  $1.5 \times 10^{-8}$  Kg m<sup>-2</sup> s<sup>-1</sup>. The corresponding maximum  $\overline{\beta}_F \approx 0.16 \times 10^{-5}$  Mm<sup>-1</sup> Sr<sup>-1</sup> nm<sup>-1</sup> is lower than most previously reported values for BBA in France (Hu et al., 2022)  $\overline{\beta}_F \approx 1.8 \times 10^{-5}$  Mm<sup>-1</sup> Sr<sup>-1</sup> nm<sup>-1</sup>; Germany (Gast et al., 2025)  $\overline{\beta}_F \approx 11 \times 10^{-5}$  Mm<sup>-1</sup> Sr<sup>-1</sup> nm<sup>-1</sup>. This study high-lights the high sensitivity of the LIF lidar: even in late April 2024 (weak ICP fire activity), it still detected a weak BBA layer over South China. Additionally, our results show that long-range BBA transport from the ICP to South China can occur even during periods of relatively weak fire activity (e.g., late April).

Biomass burning emits complex carbonaceous aerosols, which undergo further chemical transformations during atmospheric transport and result in more diverse compositions (Hodshire et al., 2019; Wang et al., 2019). Many components in BBA, such as polycyclic aromatic hydrocarbons (PAHs) and humic-like substances (HULIS), can produce fluorescence under UV excitation (Pöhlker et al., 2012; Yue et al., 2022). In Case 1, the 1.8–2.4 km layer (Fig. 4b) exhibited a spectral shoulder starting at channel 18 (central wavelength is 459.5 nm), showing the presence of fluorophores. As summarized in Table C1, fluorescence peaks in the 440–480 nm range have been reported for certain PAHs and HULIS under ~355 nm excitation, which are similar to our observed spectral feature. In contrast, previous LIF lidar observations in Europe reported BBA fluorescence peaks at 482–560 nm (Table C2), highlighting the influence of source characteristics and atmospheric processing on spectral signatures.

210

Notably, the low-altitude (0.8–1.4 km) layer in Case 1, despite lacking visually distinct fluorescence enhancement in Fig. 3, exhibited a weak spectral peak resembling that of the high-altitude (1.8–2.4 km) BBA layer. This suggests partial downward mixing of BBA and indicates that even low-intensity fires can introduce potentially hazardous fluorescent species into the lower troposphere. Future work combining LIF lidar with in situ measurements could help further investigate the mechanisms of BBA downward transport (Dajuma et al., 2020).

To further investigate BBA transport processes, we analyzed radiosonde observations from multiple sites in South China (Fig. 7a), with colors representing different observation times corresponding to the BBA transport period (Fig. 5d, the "long-range transport" stage). In Fig. 7g, the  $\overline{\beta}_F$  shows a marked increase at 1.8 km, coinciding with enhanced  $\beta_{H_2O}$  at Nanping. Although previous studies have noted fluorescence interference in water vapor Raman channels (Chouza et al., 2022; Liu et al., 2022), our measurements show the fluorescence signal was roughly an order of magnitude lower than the water vapor signal (shown in Fig. 4b), indicating that fluorescence interference is unlikely to be the primary cause of the observed  $\beta_{H_2O}$  enhancement. The ERA5 RH profile (Fig. 7h) additionally verifies the presence of a high-humidity layer. As shown in Fig. 5d, HYSPLIT back trajectories indicate that the observed BBA layer was originated from coastal fire sources and transported inland by onshore flow. Furthermore, radiosonde data (Fig. 7b–f) reveal that the BBA was co-transported with water vapor, a feature consistent with previous lidar and in situ observations (Kim et al., 2009; Fadnavis et al., 2013; Pistone et al., 2021; Chavan et al., 2021; Hu et al., 2022; Rubin et al., 2023; Pistone et al., 2024) and suggests possible entrainment of marine aerosols (such as sea salt) (Dang et al., 2022). Such humid, sea salt containing conditions could enhance sunlight driven reactions, accelerating secondary sulfate formation and thereby exerting important environmental impacts (Tang et al., 2023, 2025b).

## 6 Conclusion

In this study, we reported LIF lidar observations at Nanping, South China during April–May, 2024. The lidar captured a weak but distinct fluorescent layer whose spectral characteristics differ from those of urban aerosols. Source attribution indicates the layer originated from low-intensity fire sources in the ICP, and the aerosol underwent a three-stage transport process (local biomass burning, vertical lifting, and long-range transport). These observations demonstrate the high sensitivity of LIF lidar to relatively weak fires and faint fluorescence signals. Fluorescence spectral analysis reveals peaks similar to previous in situ fluorescence studies of certain hazardous organics, yet the overall spectra differ from previous LIF lidar studies, implying compositional variability linked to source or aging. Notably, we also observed co-transport of water vapor with the BBA, which may enhance aerosol processing and increase impacts in downwind regions. Future studies should aim to couple LIF lidar with coordinated in situ observations to better evaluate impacts on low-altitude air; upgrades to the lidar system, including elastic and polarization channels, are under consideration.

Figure 7. Radiosonde, lidar and reanalysis profiles for 19 April 2024 (UTC+8), Case 1. (a) Map overview of radiosonde launch site and the Nanping lidar station; marker colors and shapes indicate acquisition time; the LIF lidar measurement period is 19 April 2024, 23:19–20 April 00:05 (UTC+8). (b–f) Water vapor mixing ratio (WVMR) profiles from radiosonde measurements. (g) Water vapor Raman backscatter coefficient  $\beta_{\text{H}_2\text{O}}$  (Mm<sup>-1</sup> Sr<sup>-1</sup>) and fluorescence backscatter coefficient  $\overline{\beta}_F$  (Mm<sup>-1</sup> Sr<sup>-1</sup> nm<sup>-1</sup>) profiles. (h) ERA5 relative humidity (RH) profile for Nanping at 19 April 2024, 23:00 (UTC+8).

# 230 Appendix A: Meteorological parameters during the entire observation period

Figure A1. Temporal variations of pollution and meteorological parameters at the Nanping lidar station. Gray shaded bands indicate the time intervals of the four cases studied. (a)  $PM_{10}$  and  $PM_{2.5}$  concentrations. (b) Boundary layer height (BLH) derived from ERA5 hourly reanalysis.

Appendix B: Source attribution of Case 1 with a starting altitude of 1 km

**Figure B1.** HYSPLIT backward trajectory analysis initialized at 1 km above ground level (Nanping lidar station marked by black triangle). (a) Vertical variation of air mass altitude for the study period. (b) Horizontal (longitude-latitude) trajectories overlaid on presumed vegetation fire locations; marker colors and shapes indicate FRP levels.

# Appendix C: Fluorescence spectra of previous studies

Table C1. Previous in-stiu studies showing fluorescence signatures similar to the BBA spectrum observed in Case 1.

| Substances                   | Excitation/emission wavelength (nm) | References             |
|------------------------------|-------------------------------------|------------------------|
| Pyrene                       | 355/450–470                         |                        |
| Fluoranthene                 | 355/460–480                         | (Zhang et al., 2019)   |
| BBA (volume correction)      | 355/450–470                         |                        |
| Cluster 5 (from ambient air) | 351/440–470                         | (Pan et al., 2012)     |
| HULIS                        | 340/475 (max)                       | (Muller et al., 2008)  |
| Fresh fulvic acids           | 332–358/410–456                     | (Klapper et al., 2002) |
| Fluoranthene                 | 355/450–470                         | (Louzon et al., 2025)  |

Table C2. Other reported LIF lidar fluorescence signatures of biomass burning aerosols.

| Origin or time       | Excitation/emission wavelength (nm) | References                 |
|----------------------|-------------------------------------|----------------------------|
| East and west Canada | 355/505–518                         |                            |
| West Canada          | 355/500–545                         | (Reichardt et al., 2025)   |
| In gardening season  | 355/482–545                         |                            |
| American, Siberian   | 355/513, 560                        | (Veselovskii et al., 2025) |

Data availability. Lidar data are available upon request(zkl@nuist.edu.cn).

Author contributions. ZL processed the data and wrote the paper. DT performed the lidar measurements and contributed to improving the manuscript. TW, SY, JC, KW, and HX helped with manuscript preparation and data analysis. ZZ, JH, HH, and YW contributed to manuscript revision.

Competing interests. The authors declare that they have no conflict of interests.

Acknowledgements. The authors thank the following data sources: the NOAA Air Resources Laboratory (ARL) for providing the HYSPLIT model (https://www.ready.noaa.gov/), NASA FIRMS (https://firms.modaps.eosdis.nasa.gov/) and NASA worldview

240 (https://worldview.earthdata.nasa.gov/) for providing satellite products, NASA and PHOTON for providing AERONET (https://aeronet.gsfc.nasa.gov/) sites data (Site managers: Sumaman Buntoung and Somjet Pattarapanitchai), the University of Wyoming's weather website(https://weather.uwyo.edu/) for providing radiosonde data, ECMWF for providing ERA5 data (https://cds.climate.copernicus.eu/), NASA for providing MERRA-2 data (https://disc.gsfc.nasa.gov/), CNEMC for providing air quality data (available at https://quotsoft.net/air/), and NOAA (https://www.ncei.noaa.gov/) for providing DEM data, and authors also thank FLATICON website for providing the icons used in the graphical abstract (https://www.flaticon.com/).

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
