# Peer review of "Laser-induced fluorescence lidar detection of weak biomass burning aerosols at Nanping, South China"

_EGUsphere, 2025_

## Referee Comment (RC1)

**Comments on "Laser-induced fluorescence lidar detection of weak biomass burning aerosols at Nanping, South China" by Zhekai Li et al.**

**Summary:**

This manuscript presents laser-induced fluorescence lidar measurements at Nanping, South China. Four measurement cases in April and May 2024 are considered to discuss the fluorescence properties of urban and biomass burning aerosol (BBA) particles in the lower troposphere. Furthermore, back-trajectory analyses are used for source attribution of one observed BBA layer. By using satellite observations and radiosonde data, the authors discuss potential transport processes of the BBA from the source region to the lidar site.

**General comment:**

The main benefit of this study is the geographical location where the presented measurements were performed. While the laser-induced fluorescence technique has been mainly characterized for lidar measurements in Europe, this study applies it also to measurements of BBA and urban aerosol in Asia.

However, the aerosol classification by means of the optical properties retrieved from the lidar measurements is not yet fully convincing. Up to now, the authors only use extensive optical properties (spectral fluorescence backscatter coefficient and aerosol extinction coefficient) and argue qualitatively ("distinct fluorescence [..] despite relatively low [extinction]") that the fluorescing aerosol layer in Case 1 should be BBA. Therefore, I suggest a major revision regarding the aerosol classification and recommend to show and discuss also the spectral fluorescence **capacity** (Reichardt, 2014), which is a **quantitative** measure for the distinction between different aerosol types (Veselovskii et al., 2022).

Furthermore, only one weak BBA layer at the end of the fire season has been considered so far. In the future, more cases also during the peak fire activity should be analyzed to gain a more mature insight into the optical properties of BBA over this region.

The structure of chapter 2 could be revised as some of the section titles are not representative for the data described in the sections. For details, please refer to the specific comments below.

**Specific comments:**

- **I. 49:** "The system emitted 355 nm lasers from an Nd:YAG laser..." Please mind to be precise in the language. I assume this should be: "The system emitted a 355 nm laser beam from an Nd:YAG laser..."?
- **I. 51:** In my opinion, "...echo signal..." is not a suitable word here. Maybe use "backscattered signal" or "return signal" instead?
- I. 61: "2.2 Satellite and weather observations"

Where are weather observations introduced in this section? Furthermore, AERONET (introduced in I. 70) is neither satellite nor weather observation – it is ground-based passive aerosol remote sensing. Please consider to find a more suitable section title or restructure the sections in this chapter, as the meteorological data are introduced in Section 2.3.

- **I. 74:** "...their geographic locations are marked as red rectangles in Fig. 5e." Looking at Fig. 5e, the locations of the AERONET stations are actually marked as red **triangles**.
- I. 95: "3 Calibration and retrieval of LIF lidar"

Please be more concise: The retrieval of which LIF lidar quantity do you mean?

- **II. 101-105:** Regarding the correction of ghost contributions: Wouldn't it be better to use several "clean" cases (i.e., with only background signal) and average over them to get an average correction factor for each ghost line instead of using only one reference case?
- **I. 115:** "... where the superscripts "aero" and "mole" are quantities related to aerosols and molecules..."

I would rather use "indicate" instead of "are" here.

**I. 126:** "So the total fluorescence backscattering coefficient..." Please introduce the symbol  $\beta_{\rm F}$  already here.

**I. 141:** "...Multiple cloud layers are identifiable via high  $\alpha_{\rm L}^{\rm aero}(z)$  values in Fig. 2d, consistent with observations reported by Sugimoto et al."

What do you mean with this consistence? Did Sugimoto et al. report similar fluorescence values for clouds? Please clarify.

**II. 142-143:** "During the observation period, both  $\beta_F$  and  $\alpha_L^{aero}$  (0.8–1.4 km layer, Fig. 2d and f) closely match the local PM concentration trends (Fig. A1a)."

This is not really evident, according to the figures. It may hold for  $\alpha_L^{aero}$ , but  $\beta_F$  is very different for case 1 and case 4, although these cases show similar PM concentrations.

**I. 145:** "These lines of evidence support that low-altitude fluorescence is largely attributable to urban aerosols."

This statement sounds a bit too general to be concluded from 4 measurement cases only. If applicable, you could draw this conclusion only for your station. Anyway, there can be other fluorescing aerosol types present in the boundary layer. For example, how about pollens? Do they play a role for the boundary layer aerosol load at your station?

- Figure 3: What do the red and blue arrows stand for in the plots? Please explain.
- **I. 147:** "This upward shift is further evident in Fig. 3, where the  $\beta_F$  in the low-altitude layers decays rapidly in all cases except Case 3."

Which layers do you refer to as low-altitude here – don't you mean the boundary layers in the different cases?

**II. 148-151:** "In Case 1, a distinct fluorescence layer accompanied by enhanced water vapor was observed at ~1.8 km despite relatively low  $\alpha_L^{aero}$  (Fig. 2c, e, and f and Fig. 3). Such fluorescence enhancement is absent in the other three cases (Fig. 3). We attributed the 1.8–2.4 km layer in Case 1 to BBA transported from the ICP."

As already addressed in the general comment, this argumentation with extensive aerosol properties is too qualitative and not sufficient for aerosol classification. The fluorescence capacity should also be shown and discussed to draw this conclusion.

**II. 159-160:** "By contrast, the BBA layer (1.8–2.4 km layer in Case 1) exhibits stronger fluorescence with distinct peaks"

What do you mean with stronger fluorescence here? A higher fluorescence capacity? Then show this, please!

**I. 162:** "Instead, its mean spectrum exhibits a weak peak closely resembling the higher-altitude (1.8–2.4 km) BBA signature"

This is not clearly discernible to me. In my opinion, case 1 (0.8–1.4 km) looks more similar to case 2 (1.8–2.4 km) from the spectral structure. So, both could be urban aerosol. Even case 1 (1.8-2.4 km) does not show a "typical" smoke spectrum as seen in previous studies. Could this even be smoke mixed with some urban aerosol? Again, the fluorescence capacity could help the discussion here!

- **I. 165:** "...during 16–18 April, 2024 (Fig. 5c)." This should be Fig. 5d, shouldn't it?
- I. 173: "The BBA then underwent vertical lifting to a higher altitude (Fig. 5c)."2-3 km is not very high. Please name the height range to be more precise.
- **II. 179-180:** "On the contrary, HYSPLIT trajectories results with a starting altitude of 1 km show that the low-altitude fluorescence was not affected by the transported BBA (Fig. B1)." Thus, it could probably be urban aerosol, as discussed in the comment to I. 162.
- **II. 184-185:** "Fire activity peaks in March but declines sharply by late April (Huang et al., 2016)."

If fire activity peaks in March, it would be very interesting to show also a case from then. Are data from March available for that time?

**II. 189-190:** "This study highlights the high sensitivity of the LIF lidar: even in late April 2024 (weak ICP fire activity), it still detected a weak BBA layer over South China." The high sensitivity of LIF lidar to smoke has already been shown in detail in previous studies (Gast et al., 2025, Reichardt et al., 2025). Please account for that here.

**Typos:**

- I. 2: "... (LIF) lidar is a (powerful tool for detecting ..."
- **I. 6:**  ${}^{m}Mm^{-1}sr^{-1}nm^{-1}$ " (s in sr has to be small).
- I. 43: "...signatures that different from those of urban aerosols."
- **I. 71:** "...by the University of Lille—1, the French..."

**References:**

Gast, B., Jimenez, C., Ansmann, A., Haarig, M., Engelmann, R., Fritzsch, F., Floutsi, A. A., Griesche, H., Ohneiser, K., Hofer, J., Radenz, M., Baars, H., Seifert, P., and Wandinger, U.: Invisible aerosol layers: improved lidar detection capabilities by means of laser-induced aerosol fluorescence, Atmospheric Chemistry and Physics, 25, 3995–4011, https://doi.org/10.5194/acp-25-3995-2025, 2025.

Reichardt, J., 2014: Cloud and Aerosol Spectroscopy with Raman Lidar. J. Atmos. Oceanic Technol., 31, 1946–1963, https://doi.org/10.1175/JTECH-D-13-00188.1.

Reichardt, J., Lauermann, F., and Behrendt, O.: Fluorescence spectra of atmospheric aerosols, Atmospheric Chemistry and Physics, 25, 5857–5892, https://doi.org/10.5194/acp-25-5857-2025, 2025

Veselovskii, I., Hu, Q., Goloub, P., Podvin, T., Barchunov, B., and Korenskii, M.: Combining Mie–Raman and fluorescence observations: a step forward in aerosol classification with lidar technology, Atmos. Meas. Tech., 15, 4881–4900, https://doi.org/10.5194/amt-15-4881-2022, 2022.

---

## Author Comment (AC1)

* * *
Manuscript ID: EGUSPHERE-2025-4436
Title: Biomass burning aerosol transport from Indo-China Peninsula to South China: fluorescence lidar observation and analysis
Authors: Zhekai Li, Dawei Tang, Tianwen Wei, Saifen Yu, Jing Cai, Kenan Wu, Zhen Zhang, Jiadong Hu, Haobin Han, Yubin Wang, and Haiyun Xia
* * *
Dear editor Gerd Baumgarten and Referee #3:

On behalf of the co-authors, thank you for giving us an opportunity to address the Referee #3's concerns. We appreciate all the great efforts and constructive comments from Referee #3. We have revised the manuscript carefully according to the Referee #3's comments and suggestions. Our point-by-point responses are appended below. All changes made in the revised manuscript are marked in blue. Attached please find the revised version of the manuscript, which we would like to submit for your kind consideration. We are looking forward to hearing from you!

Best regards!
Sincerely yours,
Zhekai Li
State Key Laboratory of Climate System Prediction and Risk Management,
Nanjing University of Information Science & Technology
219 Ningliu rd. Nanjing, Jiangsu, CHINA, 210044.
* * *
**Anonymous Referee #3:**

**General comment:**

This study analyzes four cases of laser-induced fluorescence (LIF) lidar observations in South China. In Case 1, a pronounced fluorescence layer is detected and attributed to biomass burning aerosols (BBA) transported from the Indo-China Peninsula, based on HYSPLIT backward trajectories and other supporting datasets.

LIF lidar has been demonstrated as an effective tool for detecting BBA in Europe, but applications in South China remain limited. Deploying this technique in a region frequently impacted by Southeast Asian biomass burning smoke is therefore timely and valuable. The manuscript helps fill an observational gap and provides useful insights into BBA characteristics and transport pathways.

Thank you for your comment. South China is densely populated region frequently affected by transported biomass burning aerosol (BBA). Thus, this area requires sensitive laser-induced

fluorescence (LIF) lidar observations to characterize these plumes.

Overall, the manuscript provides considerable evidence to support its conclusions, but the structure of Section 4 should be revised to align with the logical flow of evidence. Additionally, future perspectives should be added to the Conclusion section to provide useful references for the follow-up studies. Please refer to the specific comments below for details.

Thank you for your valuable suggestions. Section 4 has been revised, please see our response to your comments on **Lines 149-150** below for details. The Conclusion section has been enriched, please refer to our reply to Conclusion below for details.

We sincerely thank Referee #3 for constructive comments and suggestions, which have significantly improved the quality of our manuscript. Below, we provide our point-by-point responses to the comments:

**Specific comments:**

1. Line 31: 'the' should be deleted as no specific system is referred to.

Thank you for pointing this out. 'the' in **Line 31** has been deleted.

2. Lines 30-35: Besides atmospheric observations, LIF lidar has been applied in a range of other remote sensing applications, including aquatic oil spill detection and chlorophyll fluorescence monitoring. Adding a short discussion of these applications in the Introduction may help highlight the broader applicability of LIF lidar.

We appreciate your valuable suggestion. Brief discussions and relevant references on these applications have been added to the Introduction to highlight the broader applicability of LIF lidar. In addition, a concise background on Raman lidar has been included to clarify the observational context in which atmospheric fluorescence was first reported in water vapor Raman measurements. This also provides context that many LIF lidar systems use vibrational Raman channels as a molecular reference. We have also refined and supplemented the content pertaining to ambient atmosphere detection using LIF lidar, with additional relevant references incorporated, to provide a more comprehensive overview of LIF lidar applications:

"As a well-established remote sensing technique, Raman lidar has been widely applied to atmospheric studies (Mattis et al., 2002; Reichardt et al., 2012; Baumgarten, 2010), and has also been adapted for aquatic environments (Shangguan et al., 2023b). Vibrational Raman channels detect molecular scattering and are commonly used as a molecular reference in lidar measurements (Fiedler and Baumgarten, 2024). In early 2005, Immler et al. reported an unexpected enhancement in the water vapor Raman channel signal, which they attributed to fluorescence interference from BBA (Immler et al., 2005). Since then, researchers have developed single-channel (Rao et al., 2018; Li et al., 2019; Veselovskii et al., 2020, 2021; Zhang et al., 2021; Hu et al., 2022; Veselovskii et al., 2022a, b; Jiang et al., 2024; Gast et al., 2025; Yufeng et al., 2025) and multi-channel (Sugimoto et al., 2012; Reichardt, 2014; Saito et al., 2018; Reichardt, Jens et al., 2018; Richardson et al., 2019; Reichardt et al., 2023; Wang et al., 2023; Veselovskii et al., 2023, 2024; Huang et al., 2025; Tang et al., 2025a; Veselovskii et al., 2025; Reichardt et al., 2025) laser-induced fluorescence (LIF) lidar systems. In addition to observations of the ambient atmosphere, LIF lidar has also been used in the remote sensing of released bioaerosols (Christesen et al., 1993; Simard et al., 2004; Farsund et al., 2012; Wojtanowski et al., 2015; Duschek et al., 2017; Shoshanim, 2023), aquatic oil spills (Leifer et al., 2012; Sun et al., 2023), and chlorophyll (Saito et al., 2016; Zhao et al., 2023; Shangguan et al., 2023a, 2024), highlighting its broad environmental applicability. Previous LIF lidar observations of BBA in the ambient atmosphere, generally built upon Mie-Raman lidar systems, have been conducted mainly in Europe (Reichardt, 2014; Reichardt, Jens et al., 2018; Veselovskii et al., 2020, 2022a, b; Hu et al., 2022; Veselovskii et al., 2023; Reichardt et al., 2023; Veselovskii et al., 2024, 2025; Gast et al., 2025; Reichardt et al., 2025) and have shown that these systems provide high detection sensitivity (Gast et al., 2025; Reichardt et al., 2025). Within these European observations, distinct fluorescent layers and characteristic BBA spectra have been reported, and these layers often originated from long-range transport of BBA from strong fires in North America or Russia. However, BBA fluorescence spectra can vary substantially across locations and cases (Reichardt et al., 2025). Consequently, further observations in diverse regions and under weak fire conditions are warranted, particularly in areas with high biomass burning emission potential and population density such as the ICP and South China, where LIF remote sensing observations remain limited."

3. Line 39: To be clear, 'weaker cases' should be revised to 'weaker fire cases'.

Thank you for pointing this out. The wording has been revised to improve clarity and precision.

"Consequently, further observations in diverse regions and under weak fire conditions are warranted…"

4. Line 45: Please provide a brief overview of the section contents at the end of the Introduction.

Thank you for your suggestion. The overview has been added at the end of the Introduction:

"Building on these advances, we conducted LIF lidar observations at Nanping, South China, during April–May 2024. Section 2 describes the LIF lidar configuration and the multi-source datasets used in this study. Section 3 describes the calibration and retrieval of aerosol extinction as well as fluorescence backscatter coefficients from the LIF lidar data. Time-height profiles and fluorescence spectra are presented in Sect. 4.1 and 4.2 respectively, revealing a distinct fluorescent layer. Although the overall fluorescence intensity is relatively weak, its spectral signatures differ from those of urban aerosol. To further investigate the origin of this layer, HYSPLIT backward trajectory analysis is presented in Sect. 4.3, demonstrating that the layer originates from fire sources in the Indo-China Peninsula (ICP). Section 5 compares the observed fluorescence characteristics with previous LIF lidar studies. Additional radiosonde data along the transport pathway are further incorporated in this section. By jointly analyzing radiosonde and LIF lidar data, we find that the BBA layer was transported alongside elevated water vapor, indicating a humid transport pathway. Conclusions and future implications are summarized in Sect. 6."

5. Line 59: 'BBA layer' should be revised to 'fluorescent layer', as the fluorescence attribution is presented in Section 4.

We are grateful to you for pointing this out. 'BBA layer' has been revised to 'fluorescent layer'.

6. Lines 143-144: Please consider moving the sentence pertaining to spectral characteristics to Section 4.2, as it focuses on spectral analysis.

Thank you for your suggestion. The structure of Section 4 has been reorganized, please refer to our response to the comments on **Lines. 149-150** for details.

7. Line 145: Similarly, as the spectra are a key piece of evidence for characterizing urban aerosols, the corresponding conclusion should also be moved to Section 4.2.

We agree with your valuable suggestion, please refer to our response to the comments on **Lines 149-150** for details.

8. Lines 149-150: As already mentioned in the general comment, the conclusion of BBA characterization is premature to present here. It should be presented in Section 4.3.

We appreciate your suggestion. Combined with the suggestion of Referee #2, We have reorganized the structure of Section 4 to follow the flow of our evidence. The revised Section 4 is as follows:

"4.1 Vertical profiles observed by LIF lidar

Table 2. Estimates of layer-averaged spectral fluorescence capacity ($\hat{G}_F = \frac{\overline{\beta}_F}{\alpha_L^{aero}} \cdot S$), computed over the fluorescence range 444–487.4 nm (Channels 20–14). A lidar ratio $S$ of 55 sr (typical for aged smoke) is assumed (Ansmann et al., 2021).

| Cases | $\hat{G}_F$ ($\times 10^{-6}$ nm$^{-1}$) (0.8–1.4 km) | $\hat{G}_F$ ($\times 10^{-6}$ nm$^{-1}$) (1.8–2.4 km) |
|---|---|---|
| Case 1 | – | 3.1 |
| Case 2 | 1.5 | 0.4 |
| Case 3 | 1.2 | 1.4 |
| Case 4 | 0.5 | 0.2 |

In Case 1, a distinct fluorescence layer (enhanced $\overline{\beta}_F$) accompanied by enhanced water vapor was observed at ∼ 1.8 km despite relatively low $\alpha_L^{aero}$ (Fig. 2c, e, and f and Fig. 3). This enhancement is not observed in the other three cases (Fig. 3). To further analyze the fluorescence characterization, we use quantitative analyses of the spectral fluorescence capacity $G_F = \frac{\overline{\beta}_F}{\beta_L}$, where $\overline{\beta}_F$ is the spectral fluorescence backscatter coefficient and $\beta_L$ is the elastic backscattering coefficient (Reichardt, 2014; Veselovskii et al., 2022b). As $\beta_L$ was not directly available in this study, we estimated $\hat{G}_F = \frac{\overline{\beta}_F}{\alpha_L^{aero}} \cdot S$ using a typical lidar ratio S ≈ 55 sr for aged smoke (Ansmann et al., 2021). To enable direct comparability with the fluorescence wavelength range (444–488

nm) from (Gast et al., 2025), we selected Channels 20–14 (444–487.4 nm) for $\hat{G}_F$ estimation. $\hat{G}_F$ values are provided in Table 2, excluding Case 1 (0.8–1.4 km): negative $\alpha_L^{aero}$ results in negative $\hat{G}_F$, which is thus omitted. Table 2 presents the highest $\hat{G}_F$ for Case 1 (1.8–2.4 km) $\approx 3.1 \times 10^{-6}$ $nm^{-1}$, which falls within the typical smoke range of $2 \times 10^{-6}$–$9 \times 10^{-6}$ $nm^{-1}$ (Gast et al., 2025) and is at least twice as high as $\hat{G}_F$ values from other layers.

**4.2 Fluorescence spectra**

The spectrum of Case 1 (1.8–2.4 km) is distinct from other aerosol spectra (Fig. 4a), with quantitative support from spectral angle mapping (SAM) analysis (Fig. 4b): the SAM angle between Case 1 (1.8–2.4 km) and Cases 2 and 3 (0.8–1.4 km, urban aerosol) is $\sim 4.9°$, notably larger than the SAM angle ($\approx 1.14°$) between Cases 2 and 3 (0.8–1.4 km) themselves. Additionally, SAM angles between Case 1 (1.8–2.4 km) and Case 1 (0.8–1.4 km), as well as between Case 1 (1.8–2.4 km) and Case 2 (0.8–1.4 km), both exceed $4°$, further confirming the spectral dissimilarity. To better constrain the aerosol source in Case 1 (1.8–2.4 km), HYSPLIT backward trajectory analysis was performed in Sect. 4.3.

**4.3 Source attribution of the fluorescence layer in Case 1**

…Considering the distinct $\overline{\beta}_F$ layer (Fig. 2f), the highest $\hat{G}_F$ ($\approx 3.1 \times 10^{-6}$ $nm^{-1}$; Table 2), the unique fluorescence spectral shape (Fig. 4a–b), and HYSPLIT backward trajectory analysis (Fig. 5d), these lines of evidence support that BBA transported from the ICP was a major contributor to the fluorescent layer observed in Case 1 (1.8–2.4 km)."

9. Lines 160-162: Please specify only the precise height ranges to avoid confusion.

Thank you for pointing this out. The height-related sentences are revised to only precise height ranges.

10. Lines 202-205: The consideration of the influence from high altitudes to low altitudes is noteworthy, but the discussion is insufficient based solely on two weak spectra. Therefore, the authors are advised to remove the relevant descriptions and focus instead on the implications for future research.

Thank you for your suggestion. We have deleted the relevant discussions. To strengthen the discussion on future research implications, we have supplemented relevant references and revised the corresponding content as follows:

"Via vertical mixing, such transported BBA may influence the near-surface atmosphere (Dajuma et al., 2020). Future observations combining LIF lidar with in-situ instrumentation such as the Wideband Integrated Bioaerosol Sensor (WIBS), which also operates on LIF principles (Tang et al., 2022), would facilitate a more in-depth investigation of the near-surface impacts exerted by transported BBA. Combined LIF lidar and WIBS measurements have recently been reported (Gidarakou et al., 2025)."

11. Line 213: "was originated" should be revised to "originated".

Thank you for pointing this out. 'was' has been deleted.

12. Lines 213-217: Please reorder the sentences for clearer logic. The description should specify that onshore flow introduces the potential for mixing between marine aerosols and BBA.

Thank you for your suggestion. The order of sentences has been reorganized:

"As shown in Fig. 5d, HYSPLIT backward trajectories indicate that the observed BBA layer originated from fire sources near coastal regions and was transported inland by onshore flow, which suggests possible entrainment of marine aerosols (such as sea salt) (Dang et al., 2022). Furthermore, radiosonde data (Fig. 7b–f) reveal that the BBA was co-transported with water vapor, a feature consistent with previous lidar and in situ observations (Kim et al., 2009; Fadnavis et al., 2013; Pistone et al., 2021; Chavan et al., 2021; Hu et al., 2022; Rubin et al., 2023; Pistone et al., 2024)."

13. Conclusion: As noted in the general comments, it is recommended to add a brief discussion of future research prospects at the end of the Conclusion to better outline potential directions for subsequent studies.

Thank you for your insightful suggestion. Combined with the suggestion of Referee #2, we have added future research implications at the end of the Conclusion section:

"March marks the peak of seasonal biomass burning across the ICP, with widespread agricultural burning (for planting preparation) and forest fires (Gautam et al., 2013; Huang et al., 2016). As South China lies downwind of the ICP, it provides a favorable setting for long-term LIF

lidar observations of transported BBA across different stages of the burning season. To improve quantitative aerosol classification, a LIF lidar system that integrates elastic scattering, depolarization, and fluorescence detection is under development. It will enable direct retrieval of spectral fluorescence capacity $G_F$ (Reichardt, 2014) and depolarization ratio — key parameters for advancing aerosol type differentiation (Veselovskii et al., 2022) and gaining deeper insights into regional BBA characteristics."

**In addition, several relevant references have been incorporated into the revised manuscript in response to the constructive comments from both Referee #2 and Referee #3:**

Ansmann, A., Wandinger, U., Riebesell, M., Weitkamp, C., and Michaelis, W.: Independent measurement of extinction and backscatter profiles in cirrus clouds by using a combined Raman elastic-backscatter lidar, Appl. Opt., 31, 7113–7131, https://doi.org/10.1364/AO.31.007113, 1992b.

Ansmann, A., Ohneiser, K., Mamouri, R.-E., Knopf, D. A., Veselovskii, I., Baars, H., Engelmann, R., Foth, A., Jimenez, C., Seifert, P., and Barja, B.: Tropospheric and stratospheric wildfire smoke profiling with lidar: mass, surface area, CCN, and INP retrieval, Atmospheric Chemistry and Physics, 21, 9779–9807, https://doi.org/10.5194/acp-21-9779-2021, 2021.

Baumgarten, G.: Doppler Rayleigh/Mie/Raman lidar for wind and temperature measurements in the middle atmosphere up to 80 km, Atmospheric Measurement Techniques, 3, 1509–1518, https://doi.org/10.5194/amt-3-1509-2010, 2010.

Christesen, S. D., Wong, A., DeSha, M. S., Merrow, C. N., Wilson, M. W., and Butler, J. C.: Ultraviolet-laser-induced fluorescence of aerosolized bacterial spores, in: Advances in Fluorescence Sensing Technology, edited by Lakowicz, J. R. and Thompson, R. B., vol. 1885, pp. 114–121, International Society for Optics and Photonics, SPIE, https://doi.org/10.1117/12.144702, 1993.

Duschek, F., Fellner, L., Gebert, F., Grnewald, K., Khntopp, A., Kraus, M., Mahnke, P., Pargmann, C., Tomaso, H., and Walter, A.: Standoff detection and classification of bacteria by multispectral laser-induced fluorescence, Advanced Optical Technologies, 6, 75–83, https://doi.org/10.1515/aot-2016-0066, 2017.

Farsund, O., Rustad, G., Kasen, I., and Haavardsholm, T. V.: Required Spectral Resolution for Bioaerosol Detection Algorithms Using Standoff Laser-Induced Fluorescence Measurements, IEEE Sensors Journal, 10, 655–661, https://doi.org/10.1109/JSEN.2009.2037794, 2010.

Farsund, Ø., Rustad, G., and Skogan, G.: Standoff detection of biological agents using laser induced fluorescence—a comparison of 294 nm and 355 nm excitation wavelengths, Biomed. Opt. Express, 3, 2964–2975, https://doi.org/10.1364/BOE.3.002964, 2012.

Fiedler, J. and Baumgarten, G.: The ALOMAR Rayleigh/Mie/Raman lidar: status after 30 years of operation, Atmospheric Measurement Techniques, 17, 5841–5859, https://doi.org/10.5194/amt-17-5841-2024, 2024.

Gidarakou, M., Papayannis, A., Gao, K., Gidarakos, P., Crouzy, B., Foskinis, R., Erb, S., Zhang, C., Lieberherr, G., Collaud Coen, M., Sikoparija, B., Kanji, Z. A., Clot, B., Calpini, B., Giagka, E., and Nenes, A.: Profiling pollen and biomass burning particles over Payerne, Switzerland using laser-induced fluorescence lidar and in situ techniques during the 2023 PERICLES campaign, EGUsphere, 2025, 1–34, https://doi.org/10.5194/egusphere-2025-2978, 2025.

Haarig, M., Engelmann, R., Baars, H., Gast, B., Althausen, D., and Ansmann, A.: Discussion of the spectral slope of the lidar ratio between 355 and 1064 nm from multiwavelength Raman lidar observations, Atmospheric Chemistry and Physics, 25, 7741–7763, https://doi.org/10.5194/acp-25-7741-2025, 2025.

Hu, Q., Goloub, P., Veselovskii, I., Podvin, T., Dubois, G., Khaykin, S., Boissière, W., Ducos, F., and Korenskiy, M.: Advanced insights into biomass burning aerosols during the 2023 Canadian wildfires from dual-site Raman and fluorescence lidar observations, EGUsphere, 2025, 1–40, https://doi.org/10.5194/egusphere-2025-5041, 2025.

Jiang, Y., Yang, H., Tan, W., Chen, S., Chen, H., Guo, P., Xu, Q., Gong, J., and Yu, Y.: Observation and Classification of Low-Altitude Haze Aerosols Using Fluorescence-Raman-Mie Polarization Polarization Lidar in Beijing during Spring 2024, Remote Sensing, 16, https://doi.org/10.3390/rs16173225, 2024.

Leifer, I., Lehr, W. J., Simecek-Beatty, D., Bradley, E., Clark, R., Dennison, P., Hu, Y., Matheson, S., Jones, C. E., Holt, B., Reif, M., Roberts, D. A., Svejkovsky, J., Swayze, G., and Wozencraft, J.: State of the art satellite and airborne marine oil spill remote sensing: Application to the BP Deepwater Horizon oil spill, Remote Sensing of Environment, 124, 185–209, https://doi.org/10.1016/j.rse.2012.03.024, 2012.

Li, B., Chen, S., Zhang, Y., Chen, H., and Guo, P.: Fluorescent aerosol observation in the lower atmosphere with an integrated fluorescence-Mie lidar, Journal of Quantitative Spectroscopy and Radiative Transfer, 227, 211–218, https://doi.org/10.1016/j.jqsrt.2019.02.019, 2019.

Mattis, I., Ansmann, A., Althausen, D., Jaenisch, V., Wandinger, U., Müller, D., Arshinov, Y. F., Bobrovnikov, S. M., and Serikov, I. B.: Relative-humidity profiling in the troposphere with a Raman lidar, Appl. Opt., 41, 6451–6462, https://doi.org/10.1364/AO.41.006451, 2002.

Reichardt, J.: Cloud and Aerosol Spectroscopy with Raman Lidar, Journal of Atmospheric and Oceanic Technology, 31, 1946–1963, https://doi.org/10.1175/JTECH-D-13-00188.1, 2014.

Reichardt, J., Wandinger, U., Klein, V., Mattis, I., Hilber, B., and Begbie, R.: RAMSES: German Meteorological Service autonomous Raman lidar for water vapor, temperature, aerosol, and cloud measurements, Appl. Opt., 51, 8111–8131, https://doi.org/10.1364/AO.51.008111, 2012.

Saito, Y., Kakuda, K., Yokoyama, M., Kubota, T., Tomida, T., and Park, H.-D.: Design and daytime performance of laser-induced fluorescence spectrum lidar for simultaneous detection of multiple components, dissolved organic matter, phycocyanin, and chlorophyll in river water, Appl. Opt., 55, 6727–6734, https://doi.org/10.1364/AO.55.006727, 2016.

Shangguan, M., Guo, Y., Liao, Z., and Lee, Z.: Sensing profiles of the volume scattering function at

180° using a single-photon oceanic fluorescence lidar, Opt. Express, 31, 40393–40410, https://doi.org/10.1364/OE.505615, 2023a.

Shangguan, M., Yang, Z., Shangguan, M., Lin, Z., Liao, Z., Guo, Y., and Liu, C.: Remote sensing oil in water with an all-fiber underwater single-photon Raman lidar, Appl. Opt., 62, 5301–5305, https://doi.org/10.1364/AO.488872, 2023b.

Shangguan, M., Guo, Y., and Liao, Z.: Shipborne single-photon fluorescence oceanic lidar: instrumentation and inversion, Opt. Express, 32, 10204–10218, https://doi.org/10.1364/OE.515477, 2024.

Shoshanim, O.: Detection of bioaerosols using hyperspectral LIF-LIDAR during S/K challenge II campaign at Dugway, Atmospheric Pollution Research, 14, 101723, https://doi.org/10.1016/j.apr.2023.101723, 2023.

Simard, J.-R., Roy, G., Mathieu, P., Larochelle, V., McFee, J., and Ho, J.: Standoff sensing of bioaerosols using intensified range-gated spectral analysis of laser-induced fluorescence, IEEE Transactions on Geoscience and Remote Sensing, 42, 865–874, https://doi.org/10.1109/TGRS.2003.823285, 2004.

Sun, L., Zhang, Y., Ouyang, C., Yin, S., Ren, X., and Fu, S.: A portable UAV-based laser-induced fluorescence lidar system for oil pollution and aquatic environment monitoring, Optics Communications, 527, 128914, https://doi.org/10.1016/j.optcom.2022.128914, 2023.

Tang, D., Wei, T., Yuan, J., Xia, H., and Dou, X.: Observation of bioaerosol transport using wideband integrated bioaerosol sensor and coherent Doppler lidar, Atmospheric Measurement Techniques, 15, 2819–2838, https://doi.org/10.5194/amt-15-2819-2022, 2022.

Veselovskii, I., Hu, Q., Ansmann, A., Goloub, P., Podvin, T., and Korenskiy, M.: Fluorescence lidar observations of wildfire smoke inside cirrus: a contribution to smoke–cirrus interaction research, Atmospheric Chemistry and Physics, 22, 5209–5221, https://doi.org/10.5194/acp-22-5209-2022, 2022a.

Veselovskii, I., Hu, Q., Goloub, P., Podvin, T., Barchunov, B., and Korenskii, M.: Combining Mie–Raman and fluorescence observations: a step forward in aerosol classification with lidar technology, Atmospheric Measurement Techniques, 15, 4881–4900, https://doi.org/10.5194/amt-15-4881-2022, 2022b.

Veselovskii, I., Hu, Q., Goloub, P., Podvin, T., Boissiere, W., Korenskiy, M., Kasianik, N., Khaykyn, S., and Miri, R.: Derivation of depolarization ratios of aerosol fluorescence and water vapor Raman backscatters from lidar measurements, Atmospheric Measurement Techniques, 17, 1023–1036, https://doi.org/10.5194/amt-17-1023-2024, 2024.

Wojtanowski, J., Zygmunt, M., Muzal, M., Knysak, P., Modzianko, A., Gawlikowski, A., Drozd, T., Kopczyski, K., Mierczyk, Z., Kaszczuk, M., Traczyk, M., Gietka, A., Piotrowski, W., Jakubaszek, M., and Ostrowski, R.: Performance verification of a LIF-LIDAR technique for stand-off detection and classification of biological agents, Optics & Laser Technology, 67, 25–32, https://doi.org/10.1016/j.optlastec.2014.08.013, 2015.

Yufeng, W., Xueqiao, X., Wei, C., Huige, D., Jingjing, L., and Dengxin, H.: Lidar-based investigation of aerosol hygroscopic growth characteristics using fluorescence capacity, Opt. Express, 33, 48560–48574, https://doi.org/10.1364/OE.578064, 2025.

Zhao, H., Zhou, Y., Wu, H., Kutser, T., Han, Y., Ma, R., Yao, Z., Zhao, H., Xu, P., Jiang, C., Gu, Q., Ma, S., Wu, L., Chen, Y., Sheng, H., Wan, X., Chen, W., Chen, X., Bai, J., Wu, L., Liu, Q., Sun, W., Yang, S., Hu, M., Liu, C., and Liu, D.: Potential of Mie-Fluorescence-Raman Lidar to Profile Chlorophyll a Concentration in Inland Waters, Environmental Science & Technology, 57, 14226–14236, https://doi.org/10.1021/acs.est.3c04212, pMID: 37713595, 2023.

---

## Author Comment (AC2)

* * *
Manuscript ID: EGUSPHERE-2025-4436
Title: Biomass burning aerosol transport from Indo-China Peninsula to South China: fluorescence lidar observation and analysis
Authors: Zhekai Li, Dawei Tang, Tianwen Wei, Saifen Yu, Jing Cai, Kenan Wu, Zhen Zhang, Jiadong Hu, Haobin Han, Yubin Wang, and Haiyun Xia
* * *
Dear editor Gerd Baumgarten and Referee #2:

On behalf of the co-authors, thank you for giving us an opportunity to address the Referee #2's concerns. We appreciate all the great efforts and constructive comments from Referee #2. We have revised the manuscript carefully according to the Referee #2's comments and suggestions. Our point-by-point responses are appended below. All changes made in the revised manuscript are marked in blue. Attached please find the revised version of the manuscript, which we would like to submit for your kind consideration. We are looking forward to hearing from you!

Best regards!
Sincerely yours,
Zhekai Li
State Key Laboratory of Climate System Prediction and Risk Management,
Nanjing University of Information Science & Technology
219 Ningliu rd. Nanjing, Jiangsu, CHINA, 210044.
* * *
**Our response to Referee #2's comments:**

**General comment:**

The main benefit of this study is the geographical location where the presented measurements were performed. While the laser-induced fluorescence technique has been mainly characterized for lidar measurements in Europe, this study applies it also to measurements of BBA and urban aerosol in Asia.

Thank you for your comment. This study follows previous laser-induced fluorescence (LIF) lidar applications that have primarily been conducted in Europe, and extends the application of this technique to observations in Asia.

However, the aerosol classification by means of the optical properties retrieved from the lidar measurements is not yet fully convincing. Up to now, the authors only use extensive optical properties (spectral fluorescence backscatter coefficient and aerosol extinction coefficient) and argue qualitatively ("distinct fluorescence [..] despite relatively low [extinction]") that the fluorescing aerosol layer in Case 1 should be BBA. Therefore, I suggest a major revision regarding the aerosol classification and recommend to show and discuss also the spectral fluorescence **capacity** (Reichardt, 2014), which is a **quantitative** measure for the distinction

between different aerosol types (Veselovskii et al., 2022).

Thank you for your valuable and constructive comment. To address this concern, we have restructured Sect. 4 to present a clearer and more systematic evidence chain, with an emphasis on quantitative analysis as recommended.

Specifically, we have added an analysis of the spectral fluorescence capacity $G_F$, defined as $G_F = \frac{\overline{\beta}_F}{\beta_L}$, where $\overline{\beta}_F$ is the spectral fluorescence backscatter coefficient and $\beta_L$ is the elastic backscatter coefficient (Reichardt, 2014; Veselovskii et al., 2022). As $\beta_L$ was not directly available in this study, we estimated $\hat{G}_F = \frac{\overline{\beta}_F}{\alpha_L^{aero}} \cdot S$ using a lidar ratio $S \approx 55$ sr (typical for aged smoke, Ansmann et al., 2021). To ensure direct comparability with the fluorescence wavelength range (444–488 nm) from Gast et al. (2025), we selected Channels 20–14 (444–487.4 nm) for $\hat{G}_F$ estimation.

In addition, to further quantify the spectral similarity, we have adopted the spectral angle mapping (SAM) algorithm (Farsund et al., 2010) using the same wavelength range. This method treats each spectrum as a vector and calculates the angular distance between vectors, with a range of 0° (identical spectral shape) to 90° (completely distinct spectral shape). Details of the BBA characterization are provided in our response to the comments on **ll. 148–151**. Furthermore, we have emphasized future instrumental improvements in the Conclusion section:

"To improve quantitative aerosol classification, a LIF lidar system that integrates elastic scattering, depolarization, and fluorescence detection is under development. It will enable direct retrieval of spectral fluorescence capacity $G_F$ (Reichardt, 2014) and depolarization ratio—key parameters for refining aerosol type differentiation (Veselovskii et al., 2022) and gaining deeper insights into regional BBA characteristics."

Furthermore, only one weak BBA layer at the end of the fire season has been considered so far. In the future, more cases also during the peak fire activity should be analyzed to gain a more mature insight into the optical properties of BBA over this region.

We agree with your insightful suggestion. The Indo-China Peninsula (ICP) exhibits seasonal fire activity, making South China (downwind area of ICP) suitable for long-term LIF lidar observations. We have therefore added future perspective in the Conclusion section:

"March marks the peak of seasonal biomass burning across the ICP, with widespread agricultural burning (for planting preparation) and forest fires (Gautam et al., 2013; Huang et al., 2016). As South China lies downwind of the ICP, it provides a favorable setting for long-term LIF lidar observations of transported BBA across different stages of the burning season."

The structure of chapter 2 could be revised as some of the section titles are not representative for the data described in the sections.

Thank you for your valuable comment. Section 2 has been revised to ensure all section titles accurately reflect their respective content. Please see our response to your comments on **l. 61** for details.

We sincerely thank Referee #2 for constructive comments and suggestions, which have significantly improved the quality of our manuscript. Below, we provide our point-by-point responses to the comments:

**Specific comments:**

**l. 49:** "*The system emitted 355 nm lasers from an Nd:YAG laser…*"
Please mind to be precise in the language. I assume this should be: "The system emitted a 355 nm laser beam from an Nd:YAG laser…"?

Thank you for your reminder. The sentence has been revised as proposed.

"The system emitted a 355 nm laser beam from an Nd:YAG laser…"

**l. 51:** In my opinion, "…echo signal…" is not a suitable word here. Maybe use "backscattered signal" or "return signal" instead?

We appreciate your suggestion. The term "echo signal" has been revised to "backscattered signal" as suggested.

**l. 61:** "*2.2 Satellite and weather observations*"
Where are weather observations introduced in this section? Furthermore, AERONET (introduced in l. 70) is neither satellite nor weather observation – it is ground-based passive aerosol remote sensing. Please consider to find a more suitable section title or restructure the sections in this chapter, as the meteorological data are introduced in Sect. 2.3.

We are grateful for your valuable comment. To address your concerns regarding the mismatched section content and titles, the subsection titles for Sect. 2.2 and 2.3 have been revised. The updated structure of Sect. 2 is as follows:

"2 Observations and data
2.1 Multi-channel LIF lidar
2.2 Satellite, radiosonde and ground-based observations
2.3 Reanalysis data and trajectory model"

**l. 74:** "*…their geographic locations are marked as red rectangles in Fig. 5e.*"

Looking at Fig. 5e, the locations of the AERONET stations are actually marked as red triangles.

Thank you for pointing this out. The term "rectangles" has been revised to "triangles".

**l. 95**: "*3 Calibration and retrieval of LIF lidar*"

Please be more concise: The retrieval of which LIF lidar quantity do you mean?

Thank you for your suggestion. To improve conciseness and clarify the specific retrieval quantities, Sect. 3 has been revised and divided into two subsections. The updated structure of the revised Sect. 3 is as follows:

"3 LIF lidar data processing

3.1 Ghost line calibration

3.2 Retrieval of aerosol extinction and fluorescence backscatter coefficients"

**ll. 101-105:** Regarding the correction of ghost contributions: Wouldn't it be better to use several "clean" cases (i.e., with only background signal) and average over them to get an average correction factor for each ghost line instead of using only one reference case?

We agree with your valuable suggestion and have revised the ghost correction procedure accordingly. Specifically, we derived channel-specific average ghost line correction coefficients from three reference spectra, each selected from a relatively clean case where the $N_2$ Raman signal maintains sufficient intensity to ensure calibration reliability. As shown in Fig. 1, the corrected spectrum of Case 4 shows minor changes (bottom dashed blue line): intensity depressions appear in the channels corresponding to ghost lines, which is attributed to the elevated correction coefficients from the averaging procedure. This variation, however, does not significantly affect the subsequent spectral analysis: the fluorescence signal is four orders of magnitude weaker than the $N_2$ Raman signal, and thus dominated by natural fluctuations rather than ghost artifacts.

"To quantify ghost contributions and standardize correction across all spectral data, we selected three reference spectra with the lowest fluorescence intensity from three cases (where the $N_2$ Raman signal maintains sufficient intensity to ensure calibration reliability). One of them is the spectrum from Case 4 (2.0–2.8 km), which is shown in Fig. 1. [...] The final channel-specific ghost correction coefficients were obtained by averaging the corresponding coefficients from the three reference spectra."

[Figure]

Figure 1. Mean fluorescence spectra for Cases 1 (1.0–1.8 km), 2 (1.0–1.8 km), and 4 (2.0–2.8 km) measured at the LIF lidar site. Line colors indicate the different cases. Solid lines show spectra before ghost-line correction and dashed lines show spectra after ghost-line correction (see legend). All the spectra are normalized by the $N_2$ Raman signal.

**l. 115:** "*...where the superscripts "aero" and "mole"* *are* *quantities related to aerosols and molecules…*"

I would rather use "indicate" instead of "are" here.

Thank you for pointing this out. The term "are" has been revised to "indicate".

**l. 126:** "So the total fluorescence backscattering coefficient…"

Please introduce the symbol $\beta_F$ already here.

Thank you for your suggestion. The symbol $\beta_F$ has been incorporated in this sentence, and the sentence has been rewritten for clarity:

"For comparison, $\beta_F$ is then normalized by the fluorescence spectral wavelength range to yield the spectral fluorescence backscatter coefficient:"

**l. 141:** "*...Multiple cloud layers are identifiable via high $\alpha_L^{aero}$ values in Fig. 2d,* *consistent with observations reported by Sugimoto et al.*"

What do you mean with this consistence? Did Sugimoto et al. report similar fluorescence values for clouds? Please clarify.

We are grateful for your comment and apologize for the ambiguity in the original wording. The "consistence" refers to the consistent observational phenomenon reported in our study and Sugimoto et al. (2012): abrupt increases in extinction coefficients at specific altitudes, which are attributed to the presence of clouds.

"In Cases 1, 3, and 4 (3–4 km), we observed abrupt enhancements in $\alpha_L^{aero}(z)$ (Fig. 2d), indicating the presence of clouds (Sugimoto et al., 2012)."

**ll. 142-143:** "*During the observation period, both $\beta_F$ and $\alpha_L^{aero}$ (0.8–1.4 km layer, Fig. 2d and f) closely match the local PM concentration trends (Fig. A1a).*"

This is not really evident, according to the figures. It may hold for $\alpha_L^{aero}$, but $\beta_F$ is very different for case 1 and case 4, although these cases show similar PM concentrations.

We appreciate your insightful comment and agree with your opinion. To clarify the relationships, the original Fig. A1 has been replaced by a summary figure showing the key quantities for the four cases, including layer-averaged $\overline{\beta}_F$ and $\alpha_L^{aero}$ (0.8–1.4 km), as well as surface RH and PM concentrations.

Regarding $\alpha_L^{aero}$, Cases 1 and 4 occurred after precipitation with RH close to saturation, favoring wet deposition and resulting in reduced aerosol loading. Consequently, lower $\alpha_L^{aero}$ (< 0.05 km$^{-1}$) are observed compared to Cases 2 and 3 (> 0.15 km$^{-1}$), in qualitative agreement with the PM concentration differences. The slightly negative layer-averaged $\alpha_L^{aero}$ in Case 1 can be attributed to retrieval uncertainties under very clean conditions, such as overestimated molecular extinction or sensitivities to temperature and pressure profiles (Ansmann et al., 1992b), a behavior also reported by Hu et al. (2025). Differences between Cases 2 and 3 are likely related to hygroscopic growth under elevated RH (Ansmann et al., 1992a; Haarig et al., 2025).

Regarding $\overline{\beta}_F$, under the assumption of minimal water-induced fluorescence quenching (Veselovskii et al., 2025a), negligible hygroscopic effects on aerosol fluorescence, and unchanged aerosol mixing state, $\overline{\beta}_F$ can be regarded as a reliable proxy for dry aerosol material concentrations (Miri et al., 2024), consistent with the dry-state nature of PM measurements. However, the opposite ordering of $\overline{\beta}_F$ and PM concentrations in Cases 2 and 3 indicates that aerosol fluorescence does not scale linearly with bulk particulate mass, reflecting differences in chemical composition and fluorescence efficiency rather than particle mass alone (Reichardt, 2014). The corresponding statements in Sect. 4.1 have been revised to:

"Figure A1 presents the relationships among averaged $\overline{\beta}_F$, $\alpha_L^{aero}$ (0.8–1.4 km), surface RH, and surface PM concentrations. In Cases 1 and 4, RH remains close to saturation following preceding precipitation, favoring efficient wet deposition and resulting in reduced aerosol loading. Consequently, lower $\alpha_L^{aero}$ (< 0.05 km$^{-1}$) are observed compared to Cases 2 and 3 (> 0.15 km$^{-1}$), which qualitatively agrees with the observed PM concentration differences. The slightly negative layer-averaged $\alpha_L^{aero}$ in Case 1 can be attributed to retrieval uncertainties under very clean conditions, a behavior also reported in previous studies (Ansmann et al., 1992b; Hu et al., 2025). In particular, hygroscopic growth under elevated RH can enhance aerosol optical extinction by

modifying particle size and refractive index (Ansmann et al., 1992a; Haarig et al., 2025). This effect likely contributes to the slightly higher $\alpha_L^{\mathrm{aero}}$ in Case 2 compared to Case 3 (RH ≈ 89.2% versus 83.6%). In contrast to aerosol extinction, fluorescence signals are expected to be much less affected by ambient humidity and hygroscopic growth. Under the assumption of minimal water-induced fluorescence quenching (Veselovskii et al., 2025a), negligible hygroscopic effects on aerosol fluorescence, and unchanged aerosol mixing state, $\overline{\beta}_F$ can be regarded as a reliable proxy for dry aerosol material concentrations (Miri et al., 2024), which is consistent with the dry-state nature of the measured PM mass concentrations. However, the relative magnitudes of $\overline{\beta}_F$ in Cases 2 and 3 still exhibit an opposite ordering compared to PM concentrations, indicating that aerosol fluorescence does not scale linearly with bulk particulate mass. This discrepancy reflects the combined influence of fluorescent particle types and concentrations, rather than particle mass alone (Reichardt, 2014)."

**Appendix A: Meteorological parameters and case comparisons**

[Figure]

Figure A1. Comparison of key parameters: averaged $\overline{\beta}_F$, $\alpha_L^{\mathrm{aero}}$ (0.8–1.4 km), surface RH, and surface PM concentrations. The values of $\alpha_L^{\mathrm{aero}}$ have been multiplied by 100 for clarity. Note that the averaged $\alpha_L^{\mathrm{aero}}$ for Case 1 is negative, which can be attributed to overestimated molecular extinction or retrieval uncertainties associated with temperature and pressure profiles (Ansmann et al., 1992b).

**l. 145:** "*These lines of evidence support that low-altitude fluorescence is largely attributable to urban aerosols.*"

This statement sounds a bit too general to be concluded from 4 measurement cases only. If applicable, you could draw this conclusion only for your station. Anyway, there can be other fluorescing aerosol types present in the boundary layer. For example, how about pollens? Do they play a role for the boundary layer aerosol load at your station?

We appreciate this insightful comment and agree that generalizing the conclusion based on four cases is risky. We have revised the statement to strictly limit the conclusion to our specific station during the observation period.

Regarding your query about pollen: In May, local tree species (e.g., pines) around the lidar site

do release pollen. However, our analysis suggests that the fluorescence signal in Cases 2 and 3 (0.8–1.4 km) is likely attributable to urban aerosols rather than pollen, supported by three converging lines of evidence. First, the observed fluorescence spectra do not exhibit the characteristic peaks typical of pollen (Saito et al., 2018), and the Spectral angle mapping (SAM) analysis shows that spectra in Cases 2 and 3 (0.8–1.4 km) are similar to each other. Both spectra exhibit decreasing intensity with increasing wavelength, consistent with reported urban aerosol spectra in the boundary layer (Veselovskii et al., 2025b). Second, the estimated fluorescence capacities $\hat{G}_F$ are low ($\leq 1.5 \times 10^{-6}$ nm$^{-1}$), consistent with previous findings that pollen exhibit higher fluorescence capacity than urban aerosol (Veselovskii et al., 2022). Thirdly, the averaged $\alpha_L^{aero}$ (0.8–1.4 km) align with local PM concentration trends, linking the optical properties to the urban pollution load. We added some discussion in the revised manuscript:

[Figure]

Figure 4. (a) Mean fluorescence spectra (normalized by N$_2$ Raman signal) derived from 600 m-thick layers for Cases 1–3. Line colors and marker shapes denote the different cases and layer altitudes (see legend). (b) SAM angle matrix for the five selected spectra (Channels 20–14, 444–487.4 nm). A SAM angle of 0° indicates identical spectral shapes, with larger angles corresponding to greater dissimilarity.

"To quantitatively analyze the spectral similarity, we adopted the spectral angle mapping (SAM) analysis (Farsund et al., 2010) using Channels 20–14 (444–487.4 nm) from the fluorescence spectra. This algorithm quantifies spectral similarity by treating spectra as vectors and calculating the vector angle, ranging from 0° (identical spectral shape) to 90° (completely distinct spectral shape). Fig. 4b shows the SAM angle matrix for the selected spectra in Fig. 4a. The low SAM angle ($\approx 1.14°$) between Cases 2 and 3 (0.8–1.4 km) indicates high spectral similarity. Both spectra exhibit decreasing intensity with increasing wavelength, consistent with reported urban aerosol spectra in

the boundary layer (Veselovskii et al., 2025b). Considering these spectral features with the low $\hat{G}_F$ ($\leq 1.5 \times 10^{-6}$ nm$^{-1}$; Table 2) and the higher averaged $\alpha_L^{aero}$ (0.8–1.4 km) in Cases 2 and 3 compared to Cases 1 and 4 (linked with PM concentration trends; Fig. A1), our results suggest that the fluorescence of Cases 2 and 3 (0.8–1.4 km) observed at our LIF lidar site was likely attributable to urban aerosol. Regarding potential biogenic interference, although local dominant tree species (e.g., pines) may release pollen in May, pollen was unlikely to be the dominant contributor to the fluorescence signal in Cases 2 and 3 (0.8–1.4 km). This is evidenced by the absence of distinct characteristic peaks in the fluorescence spectra (Saito et al., 2018) and the low $\hat{G}_F$."

**Figure 3:** What do the red and blue arrows stand for in the plots? Please explain.

Thank you for pointing this out. The red and blue arrows were intended to indicate the according axes in the figure. To avoid confusion, we have removed them.

**l. 147:** "This upward shift is further evident in Fig. 3, where the $\beta_F$ in the low-altitude layers' decays rapidly in all cases except Case 3."
Which layers do you refer to as low-altitude here – don't you mean the boundary layers in the different cases?

We appreciate you for pointing this out and have revised the term to specify a precise height range.

"This upward shift is further evident in Fig. 3, where the averaged $\overline{\beta}_F$ (0.8–1.4 km) decays rapidly in all cases except Case 3."

**ll. 148-151:** "In Case 1, a distinct fluorescence layer accompanied by enhanced water vapor was observed at ~1.8 km despite relatively low $\alpha_L^{aero}$ (Fig. 2c, e, and f and Fig. 3). Such fluorescence enhancement is absent in the other three cases (Fig. 3). We attributed the 1.8– 2.4 km layer in Case 1 to BBA transported from the ICP."
As already addressed in the general comment, this argumentation with extensive aerosol properties is too qualitative and not sufficient for aerosol classification. The fluorescence capacity should also be shown and discussed to draw this conclusion.

We appreciate your valuable suggestion and have added quantitative analyses of the spectral fluorescence capacity $G_F$, which is defined as $G_F = \frac{\overline{\beta}_F}{\beta_L}$, where $\overline{\beta}_F$ is the spectral fluorescence backscatter coefficient and $\beta_L$ is the elastic backscatter coefficient (Reichardt, 2014; Veselovskii et al., 2022). As $\beta_L$ was not directly available in this study, we estimated $\hat{G}_F = \frac{\overline{\beta}_F}{\alpha_L^{aero}} \cdot S$ using a lidar ratio $S \approx 55$ sr (typical for aged smoke, Ansmann et al., 2021). To ensure direct comparability with the fluorescence wavelength range (444–488 nm) from Gast et al. (2025), we selected

Channels 20–14 (444–487.4 nm) for $\hat{G}_F$ estimation. $\hat{G}_F$ values are provided in Table 2, with Case 1 (0.8–1.4 nm) excluded: negative $\alpha_L^{aero}$ results in negative $\hat{G}_F$, which is not presented herein. Additionally, Sect. 4 has been restructured to align with the logical flow of evidence presentation. Four lines of evidence suggest that BBA transported from the Indo-China Peninsula (ICP) serves as a major contributor to the enhanced fluorescence layer observed in Case 1 (1.8–2.4 km):

(1) A distinct fluorescence layer (enhanced $\overline{\beta}_F$) was observed at ~ 1.8 km, absent in the other three cases (Fig. 2c, f and Fig. 3);

(2) $\hat{G}_F$ of Case 1 (1.8–2.4 km) is $\approx 3.1 \times 10^{-6}$ nm$^{-1}$, which falls within the typical range for aged smoke ($2 \times 10^{-6}$–$9 \times 10^{-6}$ nm$^{-1}$; Gast et al., 2025) and is at least twice as high as the corresponding $\hat{G}_F$ values from other layers;

(3) Spectral angle mapping (SAM) analysis shows the spectrum of Case 1 (1.8–2.4 km) is distinct from the urban aerosol spectra (Cases 2–3, 0.8–1.4 km): SAM angles between Case 1 and the two urban aerosol spectra are ~ 4.9°, while the SAM angle between the two urban aerosol spectra is $\approx 1.14$° (as shown in Fig. 4b above);

(4) HYSPLIT backward trajectories at 2.1 km indicate the air mass originated from fire points in the ICP (Fig. 5d).

The revised content of Sect. 4 regarding BBA characterization is as follows:

"4.1 Vertical profiles observed by LIF lidar

Table 2. Estimates of layer-averaged spectral fluorescence capacity ($\hat{G}_F = \frac{\overline{\beta}_F}{\alpha_L^{aero}} \cdot S$), computed over the fluorescence range 444–487.4 nm (Channels 20–14). A lidar ratio $S$ of 55 sr (typical for aged smoke) is assumed (Ansmann et al., 2021).

| Cases | $\hat{G}_F$ ($\times 10^{-6}$ nm$^{-1}$) (0.8–1.4 km) | $\hat{G}_F$ ($\times 10^{-6}$ nm$^{-1}$) (1.8–2.4 km) |
|---|---|---|
| Case 1 | – | 3.1 |
| Case 2 | 1.5 | 0.4 |
| Case 3 | 1.2 | 1.4 |
| Case 4 | 0.5 | 0.2 |

In Case 1, a distinct fluorescent layer (enhanced $\overline{\beta}_F$) accompanied by enhanced water vapor was observed at ~ 1.8 km despite relatively low $\alpha_L^{aero}$ (Fig. 2c, e, and f and Fig. 3). This enhancement is not observed in the other three cases (Fig. 3). To further analyze the fluorescence characterization, we use quantitative analyses of the spectral fluorescence capacity $G_F = \frac{\overline{\beta}_F}{\beta_L}$, where $\overline{\beta}_F$ is the spectral fluorescence backscatter coefficient and $\beta_L$ is the elastic backscatter coefficient (Reichardt, 2014; Veselovskii et al., 2022b). As $\beta_L$ was not directly available in this study, we

estimated $\hat{G}_F = \frac{\overline{\beta_F}}{\alpha_L^{aero}} \cdot S$ using a typical lidar ratio $S \approx 55$ sr for aged smoke (Ansmann et al., 2021). To enable direct comparability with the fluorescence wavelength range (444–488 nm) from (Gast et al., 2025), we selected Channels 20–14 (444–487.4 nm) for $\hat{G}_F$ estimation. $\hat{G}_F$ values are provided in Table 2, excluding Case 1 (0.8–1.4 nm): negative $\alpha_L^{aero}$ results in negative $\hat{G}_F$, which is thus omitted. Table 2 presents the highest $\hat{G}_F$ for Case 1 (1.8–2.4 km) $\approx 3.1 \times 10^{-6}$ nm$^{-1}$, which falls within the typical smoke range of $2\times10^{-6}$–$9\times10^{-6}$ nm$^{-1}$ (Gast et al., 2025) and is at least twice as high as $\hat{G}_F$ values from other layers.

**4.2 Fluorescence spectra**

The spectrum of Case 1 (1.8–2.4 km) is distinct from other aerosol spectra (Fig. 4a), with quantitative support from spectral angle mapping (SAM) analysis (Fig. 4b): the SAM angle between Case 1 (1.8–2.4 km) and Cases 2 and 3 (0.8–1.4 km, urban aerosol) is $\sim 4.9°$, notably larger than the SAM angle ($\approx 1.14°$) between Cases 2 and 3 (0.8–1.4 km) themselves. Additionally, SAM angles between Case 1 (1.8–2.4 km) and Case 1 (0.8–1.4 km), as well as between Case 1 (1.8–2.4 km) and Case 2 (0.8–1.4 km), both exceed $4°$, further confirming the spectral dissimilarity. To better constrain the aerosol source in Case 1 (1.8–2.4 km), HYSPLIT backward trajectory analysis was performed in Sect. 4.3.

**4.3 Source attribution of the fluorescence layer in Case 1**

…Considering the distinct $\overline{\beta}_F$ layer (Fig. 2f), the highest $\hat{G}_F$ ($\approx 3.1 \times 10^{-6}$ nm$^{-1}$; Table 2), the unique fluorescence spectral shape (Fig. 4a–b), and HYSPLIT backward trajectory analysis (Fig. 5d), these lines of evidence support that BBA transported from the ICP was a major contributor to the fluorescent layer observed in Case 1 (1.8–2.4 km)."

**ll. 159-160:** "By contrast, the BBA layer (1.8–2.4 km layer in Case 1) exhibits stronger fluorescence with distinct peaks"
What do you mean with stronger fluorescence here? A higher fluorescence capacity? Then show this, please!

Thank you for pointing this out. The original phrase "stronger fluorescence" here refers to the fluorescence intensity normalized by the $N_2$ Raman signal. We agree that fluorescence capacity is a more physically meaningful metric here. And we have included a quantitative analysis of fluorescence capacity in Sect. 4.1 to demonstrate the properties of the BBA layer explicitly. The original vague sentence (**ll. 159-160**) has been removed to avoid confusion.

**l. 162:** "Instead, its mean spectrum exhibits a weak peak closely resembling the higher altitude (1.8–2.4 km) BBA signature"
This is not clearly discernible to me. In my opinion, case 1 (0.8–1.4 km) looks more similar to case

2 (1.8–2.4 km) from the spectral structure. So, both could be urban aerosol. Even case 1 (1.8-2.4 km) does not show a "typical" smoke spectrum as seen in previous studies. Could this even be smoke mixed with some urban aerosol? Again, the fluorescence capacity could help the discussion here!

We appreciate your valuable comment. To quantitatively assess spectral similarity, we adopted the Spectral Angle Mapping (SAM) method (Farsund et al., 2010). As shown in Fig. 4b, the SAM results strongly support your assessment: the SAM angle between Case 1 (0.8–1.4 km) and Case 2 (1.8–2.4 km) is $\approx 1.36°$, while the angle between the two Case 1 layers is $\approx 4.22°$. This confirms that the spectrum of Case 1 (0.8–1.4 km) is indeed more similar to that of Case 2 (1.8–2.4 km).

Consequently, we agree that the lower layer is likely dominated by urban aerosol. Due to the low fluorescence intensity and lack of additional observational constraints, we have removed the speculative discussion regarding the downward mixing of BBA to avoid over-interpretation.

Regarding the high-altitude layer (Case 1, 1.8–2.4 km), we agree that mixing with urban aerosol cannot be excluded given the long-range transport of BBA. However, as noted in our response to your comments on **ll. 148–151**, the estimated fluorescence capacity $\hat{G}_F$ is approximately $3.1 \times 10^{-6}$ $nm^{-1}$. This value falls within the typical smoke range reported by Gast et al. (2025), supporting the conclusion that BBA transported from the ICP was a major contributor to this layer. We have rewritten the relevant discussion in Sect. 5 to explicitly address the spectral features and the likelihood of mixing:

"For the spectrum of Case 1 (1.8–2.4 km), the relatively weak peak intensity is probably attributable to low fluorescence signal intensity (which is over two orders of magnitude lower than the $N_2$ Raman signal) and mixing with urban aerosol due to the long-range transport, while the peak wavelength discrepancy relative to previous studies is likely due to distinct fire sources."

**l. 165:** "…during 16–18 April, 2024 (Fig. 5c)."
This should be Fig. 5d, shouldn't it?

We appreciate you for pointing this out. Fig. 5c has been corrected to Fig. 5d.

**l. 173:** "The BBA then underwent vertical lifting to a higher altitude (Fig. 5c)."
2-3 km is not very high. Please name the height range to be more precise.

Thank you for your suggestion. We have revised it to a more precise height range:

"The BBA then underwent vertical lifting to the 2–3 km altitude range (Fig. 5c)."

**ll. 179-180:** "On the contrary, HYSPLIT trajectories results with a starting altitude of 1 km show that the low-altitude fluorescence was not affected by the transported BBA (Fig. B1)."
Thus, it could probably be urban aerosol, as discussed in the comment to l. 162.

Thank you for pointing this out. Please see our response to your comments on **l. 162** for the revision.

**ll. 184-185:** "Fire activity peaks in March but declines sharply by late April (Huang et al., 2016)." If fire activity peaks in March, it would be very interesting to show also a case from then. Are data from March available for that time?

Thank you for this insightful suggestion. We fully concur with your view. Unfortunately, no March data were available for the present study. To address this data gap in future research, we have incorporated relevant experimental outlooks at the end of the Conclusion section, as follows:

"March marks the peak of seasonal biomass burning across the ICP, with widespread agricultural burning (for planting preparation) and forest fires (Gautam et al., 2013; Huang et al., 2016). As South China lies downwind of the ICP, it provides a favorable setting for long-term LIF lidar observations of transported BBA across different stages of the burning season. To improve quantitative aerosol classification, a LIF lidar system that integrates elastic scattering, depolarization, and fluorescence detection is under development. It will enable direct retrieval of spectral fluorescence capacity $G_F$ (Reichardt, 2014) and depolarization ratio — key parameters for advancing aerosol type differentiation (Veselovskii et al., 2022) and gaining deeper insights into regional BBA characteristics."

**ll. 189-190:** "This study highlights the high sensitivity of the LIF lidar: even in late April 2024 (weak ICP fire activity), it still detected a weak BBA layer over South China." The high sensitivity of LIF lidar to smoke has already been shown in detail in previous studies (Gast et al., 2025, Reichardt et al., 2025). Please account for that here.

Thank you for providing this key reference information. We have revised the sentences in Sect. 5 to incorporate references to previous studies on LIF lidar's smoke sensitivity, and we have also emphasized this aspect in the Introduction section:

**(In the revised Sect. 5):** "The detected fluorescent layer was relatively weak, with a fluorescence signal intensity more than two orders of magnitude lower than the $N_2$ Raman signal intensity. The maximum $\overline{\beta}_F \approx 0.16 \times 10^{-5}$ $Mm^{-1}$ $sr^{-1}$ $nm^{-1}$ observed in Case 1 (1.8–2.4 km) lies at the lower end of the range of $\overline{\beta}_F$ values reported for BBA in France (Hu et al., 2022), Germany (Gast et al., 2025; Reichardt et al., 2025), and Russia (Veselovskii et al., 2025). Consistent with the high sensitivity of LIF lidar reported in previous studies (Gast et al., 2025; Reichardt et al., 2025), our results show that weak, long-range transported BBA from the ICP can be observed over South China during periods of relatively weak fire activity (e.g., late April)."

**(In the revised Introduction Section):** "…and have demonstrated high detection sensitivity

(Gast et al., 2025, Reichardt et al., 2025)."

**Typos:**

**l. 2:** "… (LIF) lidar is a (powerful tool for detecting …"

**l. 6:** "Mm$^{-1}$**s**r$^{-1}$nm$^{-1}$" (s in sr has to be small).

**l. 43:** "…signatures that differ from those of urban aerosols."

**l. 71:** "…by the University of Lille, the French…"

    Thank you for your careful comments. All the listed typos have been corrected in the revised manuscript. In addition, we have reviewed the entire manuscript and made further language improvements.

**In addition, several relevant references have been incorporated into the revised manuscript in response to the constructive comments from both Referee #2 and Referee #3:**

Ansmann, A., Wandinger, U., Riebesell, M., Weitkamp, C., and Michaelis, W.: Independent measurement of extinction and backscatter profiles in cirrus clouds by using a combined Raman elastic-backscatter lidar, Appl. Opt., 31, 7113–7131, https://doi.org/10.1364/AO.31.007113, 1992b.

Ansmann, A., Ohneiser, K., Mamouri, R.-E., Knopf, D. A., Veselovskii, I., Baars, H., Engelmann, R., Foth, A., Jimenez, C., Seifert, P., and Barja, B.: Tropospheric and stratospheric wildfire smoke profiling with lidar: mass, surface area, CCN, and INP retrieval, Atmospheric Chemistry and Physics, 21, 9779–9807, https://doi.org/10.5194/acp-21-9779-2021, 2021.

Baumgarten, G.: Doppler Rayleigh/Mie/Raman lidar for wind and temperature measurements in the middle atmosphere up to 80 km, Atmospheric Measurement Techniques, 3, 1509–1518, https://doi.org/10.5194/amt-3-1509-2010, 2010.

Christesen, S. D., Wong, A., DeSha, M. S., Merrow, C. N., Wilson, M. W., and Butler, J. C.: Ultraviolet-laser-induced fluorescence of aerosolized bacterial spores, in: Advances in Fluorescence Sensing Technology, edited by Lakowicz, J. R. and Thompson, R. B., vol. 1885, pp. 114–121, International Society for Optics and Photonics, SPIE, https://doi.org/10.1117/12.144702, 1993.

Duschek, F., Fellner, L., Gebert, F., Grnewald, K., Khntopp, A., Kraus, M., Mahnke, P., Pargmann, C., Tomaso, H., and Walter, A.: Standoff detection and classification of bacteria by multispectral laser-induced fluorescence, Advanced Optical Technologies, 6, 75–83, https://doi.org/10.1515/aot-2016-0066, 2017.

Farsund, O., Rustad, G., Kasen, I., and Haavardsholm, T. V.: Required Spectral Resolution for Bioaerosol Detection Algorithms Using Standoff Laser-Induced Fluorescence Measurements, IEEE Sensors Journal, 10, 655–661, https://doi.org/10.1109/JSEN.2009.2037794, 2010.

Farsund, Ø., Rustad, G., and Skogan, G.: Standoff detection of biological agents using laser induced fluorescence—a comparison of 294 nm and 355 nm excitation wavelengths, Biomed. Opt.

Express, 3, 2964–2975, https://doi.org/10.1364/BOE.3.002964, 2012.

Fiedler, J. and Baumgarten, G.: The ALOMAR Rayleigh/Mie/Raman lidar: status after 30 years of operation, Atmospheric Measurement Techniques, 17, 5841–5859, https://doi.org/10.5194/amt-17-5841-2024, 2024.

Gidarakou, M., Papayannis, A., Gao, K., Gidarakos, P., Crouzy, B., Foskinis, R., Erb, S., Zhang, C., Lieberherr, G., Collaud Coen, M., Sikoparija, B., Kanji, Z. A., Clot, B., Calpini, B., Giagka, E., and Nenes, A.: Profiling pollen and biomass burning particles over Payerne, Switzerland using laser-induced fluorescence lidar and in situ techniques during the 2023 PERICLES campaign, EGUsphere, 2025, 1–34, https://doi.org/10.5194/egusphere-2025-2978, 2025.

Haarig, M., Engelmann, R., Baars, H., Gast, B., Althausen, D., and Ansmann, A.: Discussion of the spectral slope of the lidar ratio between 355 and 1064 nm from multiwavelength Raman lidar observations, Atmospheric Chemistry and Physics, 25, 7741–7763, https://doi.org/10.5194/acp-25-7741-2025, 2025.

Hu, Q., Goloub, P., Veselovskii, I., Podvin, T., Dubois, G., Khaykin, S., Boissière, W., Ducos, F., and Korenskiy, M.: Advanced insights into biomass burning aerosols during the 2023 Canadian wildfires from dual-site Raman and fluorescence lidar observations, EGUsphere, 2025, 1–40, https://doi.org/10.5194/egusphere-2025-5041, 2025.

Jiang, Y., Yang, H., Tan, W., Chen, S., Chen, H., Guo, P., Xu, Q., Gong, J., and Yu, Y.: Observation and Classification of Low-Altitude Haze Aerosols Using Fluorescence-Raman-Mie Polarization Polarization Lidar in Beijing during Spring 2024, Remote Sensing, 16, https://doi.org/10.3390/rs16173225, 2024.

Leifer, I., Lehr, W. J., Simecek-Beatty, D., Bradley, E., Clark, R., Dennison, P., Hu, Y., Matheson, S., Jones, C. E., Holt, B., Reif, M., Roberts, D. A., Svejkovsky, J., Swayze, G., and Wozencraft, J.: State of the art satellite and airborne marine oil spill remote sensing: Application to the BP Deepwater Horizon oil spill, Remote Sensing of Environment, 124, 185–209, https://doi.org/10.1016/j.rse.2012.03.024, 2012.

Li, B., Chen, S., Zhang, Y., Chen, H., and Guo, P.: Fluorescent aerosol observation in the lower atmosphere with an integrated fluorescence-Mie lidar, Journal of Quantitative Spectroscopy and Radiative Transfer, 227, 211–218, https://doi.org/10.1016/j.jqsrt.2019.02.019, 2019.

Mattis, I., Ansmann, A., Althausen, D., Jaenisch, V., Wandinger, U., Müller, D., Arshinov, Y. F., Bobrovnikov, S. M., and Serikov, I. B.: Relative-humidity profiling in the troposphere with a Raman lidar, Appl. Opt., 41, 6451–6462, https://doi.org/10.1364/AO.41.006451, 2002.

Reichardt, J.: Cloud and Aerosol Spectroscopy with Raman Lidar, Journal of Atmospheric and Oceanic Technology, 31, 1946–1963, https://doi.org/10.1175/JTECH-D-13-00188.1, 2014.

Reichardt, J., Wandinger, U., Klein, V., Mattis, I., Hilber, B., and Begbie, R.: RAMSES: German Meteorological Service autonomous Raman lidar for water vapor, temperature, aerosol, and cloud measurements, Appl. Opt., 51, 8111–8131, https://doi.org/10.1364/AO.51.008111, 2012.

Saito, Y., Kakuda, K., Yokoyama, M., Kubota, T., Tomida, T., and Park, H.-D.: Design and daytime

performance of laser-induced fluorescence spectrum lidar for simultaneous detection of multiple components, dissolved organic matter, phycocyanin, and chlorophyll in river water, Appl. Opt., 55, 6727–6734, https://doi.org/10.1364/AO.55.006727, 2016.

Shangguan, M., Guo, Y., Liao, Z., and Lee, Z.: Sensing profiles of the volume scattering function at 180° using a single-photon oceanic fluorescence lidar, Opt. Express, 31, 40393–40410, https://doi.org/10.1364/OE.505615, 2023a.

Shangguan, M., Yang, Z., Shangguan, M., Lin, Z., Liao, Z., Guo, Y., and Liu, C.: Remote sensing oil in water with an all-fiber underwater single-photon Raman lidar, Appl. Opt., 62, 5301–5305, https://doi.org/10.1364/AO.488872, 2023b.

Shangguan, M., Guo, Y., and Liao, Z.: Shipborne single-photon fluorescence oceanic lidar: instrumentation and inversion, Opt. Express, 32, 10204–10218, https://doi.org/10.1364/OE.515477, 2024.

Shoshanim, O.: Detection of bioaerosols using hyperspectral LIF-LIDAR during S/K challenge II campaign at Dugway, Atmospheric Pollution Research, 14, 101723, https://doi.org/10.1016/j.apr.2023.101723, 2023.

Simard, J.-R., Roy, G., Mathieu, P., Larochelle, V., McFee, J., and Ho, J.: Standoff sensing of bioaerosols using intensified range-gated spectral analysis of laser-induced fluorescence, IEEE Transactions on Geoscience and Remote Sensing, 42, 865–874, https://doi.org/10.1109/TGRS.2003.823285, 2004.

Sun, L., Zhang, Y., Ouyang, C., Yin, S., Ren, X., and Fu, S.: A portable UAV-based laser-induced fluorescence lidar system for oil pollution and aquatic environment monitoring, Optics Communications, 527, 128914, https://doi.org/10.1016/j.optcom.2022.128914, 2023.

Tang, D., Wei, T., Yuan, J., Xia, H., and Dou, X.: Observation of bioaerosol transport using wideband integrated bioaerosol sensor and coherent Doppler lidar, Atmospheric Measurement Techniques, 15, 2819–2838, https://doi.org/10.5194/amt-15-2819-2022, 2022.

Veselovskii, I., Hu, Q., Ansmann, A., Goloub, P., Podvin, T., and Korenskiy, M.: Fluorescence lidar observations of wildfire smoke inside cirrus: a contribution to smoke–cirrus interaction research, Atmospheric Chemistry and Physics, 22, 5209–5221, https://doi.org/10.5194/acp-22-5209-2022, 2022a.

Veselovskii, I., Hu, Q., Goloub, P., Podvin, T., Barchunov, B., and Korenskii, M.: Combining Mie–Raman and fluorescence observations: a step forward in aerosol classification with lidar technology, Atmospheric Measurement Techniques, 15, 4881–4900, https://doi.org/10.5194/amt-15-4881-2022, 2022b.

Veselovskii, I., Hu, Q., Goloub, P., Podvin, T., Boissiere, W., Korenskiy, M., Kasianik, N., Khaykyn, S., and Miri, R.: Derivation of depolarization ratios of aerosol fluorescence and water vapor Raman backscatters from lidar measurements, Atmospheric Measurement Techniques, 17, 1023–1036, https://doi.org/10.5194/amt-17-1023-2024, 2024.

Wojtanowski, J., Zygmunt, M., Muzal, M., Knysak, P., Modzianko, A., Gawlikowski, A., Drozd, T.,

Kopczyski, K., Mierczyk, Z., Kaszczuk, M., Traczyk, M., Gietka, A., Piotrowski, W., Jakubaszek, M., and Ostrowski, R.: Performance verification of a LIF-LIDAR technique for stand-off detection and classification of biological agents, Optics & Laser Technology, 67, 25–32, https://doi.org/10.1016/j.optlastec.2014.08.013, 2015.

Yufeng, W., Xueqiao, X., Wei, C., Huige, D., Jingjing, L., and Dengxin, H.: Lidar-based investigation of aerosol hygroscopic growth characteristics using fluorescence capacity, Opt. Express, 33, 48560–48574, https://doi.org/10.1364/OE.578064, 2025.

Zhao, H., Zhou, Y., Wu, H., Kutser, T., Han, Y., Ma, R., Yao, Z., Zhao, H., Xu, P., Jiang, C., Gu, Q., Ma, S., Wu, L., Chen, Y., Sheng, H., Wan, X., Chen, W., Chen, X., Bai, J., Wu, L., Liu, Q., Sun, W., Yang, S., Hu, M., Liu, C., and Liu, D.: Potential of Mie-Fluorescence-Raman Lidar to Profile Chlorophyll a Concentration in Inland Waters, Environmental Science & Technology, 57, 14226–14236, https://doi.org/10.1021/acs.est.3c04212, pMID: 37713595, 2023.